# DuoShapley: Adaptive and Scalable Shapley Value Approximation for Federated Learning

**Mengwei Yang**                                                                *mengwey@uci.edu*
*Department of Electrical Engineering and Computer Science*
*University of California, Irvine*

**Baturalp Buyukates**                                                          *b.buyukates@bham.ac.uk*
*School of Computer Science*
*University of Birmingham*

**Yanning Shen**                                                                *yannings@uci.edu*
*Department of Electrical Engineering and Computer Science*
*University of California, Irvine*

**Athina Markopoulou**                                                          *athina@uci.edu*
*Department of Electrical Engineering and Computer Science*
*University of California, Irvine*

**Reviewed on OpenReview:** *https://openreview.net/forum?id=zjgZFNEEHn*

## Abstract

Federated Learning (FL) enables collaborative model training across decentralized users while preserving data privacy, but it also raises a fundamental challenge: how to efficiently and reliably quantify individual user contributions to the global model. The Shapley value (SV) provides a principled game-theoretic framework for contribution valuation, yet its exact computation is prohibitively expensive in realistic FL systems. Existing SV approximation methods face a trade-off between scalability and estimation fidelity, particularly under heterogeneous data distributions. In this work, we propose *DuoShapley*, an efficient and adaptive SV approximation tailored to large-scale FL that adaptively balances two complementary orders: Solo, capturing individual contributions, and Leave-One-Out (LOO), capturing marginal contributions relative to the full coalition. By adaptively weighting them during training based on the alignment between local and global model updates, DuoShapley achieves both computational efficiency and accurate contribution valuation across diverse FL scenarios, from independent and identically distributed (IID) to non-IID. Beyond contribution measurement, DuoShapley enables downstream applications such as robust user selection in the presence of users with noisy data, by prioritizing users with high estimated contributions. Such selective participation leads to enhanced robustness to noisy and low-quality updates, and reduced communication overhead. Extensive experiments show that DuoShapley is both computationally efficient and effective across diverse data distributions. Hence, DuoShapley provides a practical and scalable solution for evaluating and leveraging user contributions in FL.

## 1 Introduction

Federated Learning (FL) has emerged as a widely adopted paradigm for training machine learning models over decentralized data (McMahan et al., 2017), improving over traditional centralized training approaches in various aspects, including privacy, communication, and computation efficiency. In FL, users collaboratively train a shared global model by exchanging model updates rather than raw data, reducing privacy risks and

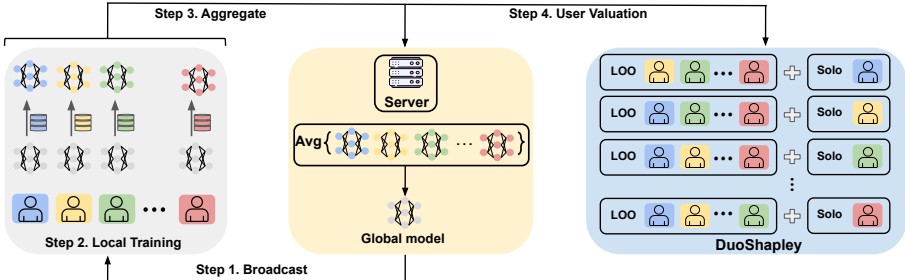

Figure 1: **Overview of the DuoShapley framework.** The server first broadcasts the global model to all users. Each user then performs local training and sends back model updates to the server. Finally, the server aggregates these updates and estimates each user's contribution by combining Solo and LOO evaluations.

communication overhead compared to centralized learning. At each iteration, the server updates the global model by aggregating model updates submitted by each user and then broadcasts the updated global model to all users (Chen & Vikalo, 2024).

Despite these advantages, FL introduces a fundamental system-level challenge: *how to efficiently measure each user's contribution to the overall learning process.* Reliable contribution valuation is essential for fair incentive allocation (Murhekar et al., 2024; Li et al., 2024a; Buyukates et al., 2023), enabling reward mechanisms such as monetary compensation or prioritized access to the final model (Wang et al., 2019). Incentive mechanisms are also essential for sustaining user engagement and collaboration (Pan et al., 2024). Without principled contribution estimates, FL systems risk under-incentivizing high-quality participants while remaining vulnerable to low-quality or harmful updates, see e.g. (Liu et al., 2022b; Cho et al., 2022; Lu et al., 2024; Allouah et al., 2024; Chen et al., 2024b; Xu et al., 2024).

The Shapley value (SV) offers a principled solution to contribution valuation by fairly attributing the global utility of a coalition to individual participants. As a result, SV-based methods have been widely studied in FL for incentive design and user selection. However, computing SV is prohibitively expensive even in moderately sized FL systems because it requires evaluating an exponential number of user coalitions. To address this challenge, prior work has proposed a variety of approximation methods, including Guided Truncation Gradient Shapley (GTG) (Liu et al., 2022a) and Truncated Monte Carlo Shapley (TMC) (Ghorbani & Zou, 2019). While effective in small-scale settings, these approaches remain computationally expensive and scale poorly as the number of users increases.

A key observation underlying this work is that scalability constraints in FL effectively restrict feasible Shapley approximations to very low-order coalitions. In particular, coalition orders beyond individual users or the full coalition quickly become impractical as the system scales. This motivates a closer examination of two extremal yet efficient coalition orders: Solo[1]: which evaluates users independently, and Leave-One-Out (LOO): which measures the marginal impact of removing a user from the full coalition. Both scale linearly with the number of users and are therefore well-suited for large-scale FL.

However, these two orders exhibit complementary limitations. Solo performs well in homogeneous (IID) settings, where user data distributions are similar. However, as distributions become more heterogeneous (non-IID) (Chen & Vikalo, 2024), its reliability declines. When a user's data differs substantially from others, its standalone performance appears limited, despite providing valuable complementary information. In such cases, Solo tends to underestimate the user's true impact, as it fails to account for collaborative interactions among users. LOO, on the other hand, evaluates each user's contribution by measuring the performance drop when the user is removed, thus capturing the influence of all other users collectively. However, it lacks resolution in homogeneous settings, where removing a single user has limited effect on the global model.

In this paper, we propose DuoShapley, an efficient Shapley value approximation that adaptively balances Solo and LOO contributions during training. By leveraging cosine similarity between local and global

---

[1]We refer to the coalition Order-1 as Solo throughout the paper, which evaluates the contribution of each user independently.

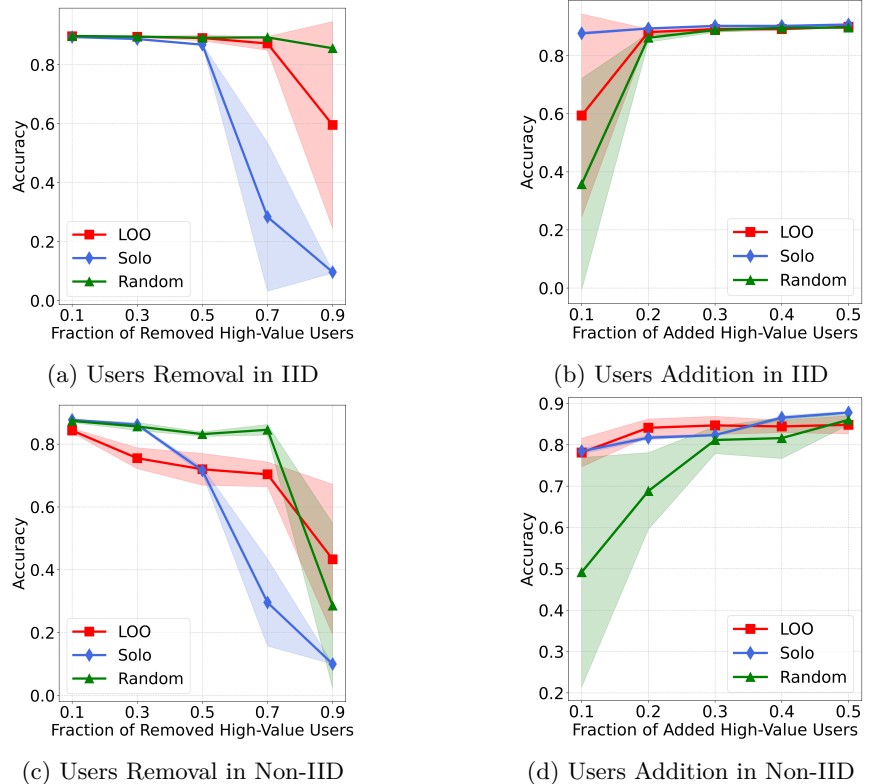

(a) Users Removal in IID       (b) Users Addition in IID

(c) Users Removal in Non-IID       (d) Users Addition in Non-IID

Figure 2: **User removal and addition under IID and non-IID settings.** We evaluate the effectiveness of efficient coalition orders, Solo, and the LOO, for estimating user contributions, with Random included as a baseline. The analysis is conducted by adding or removing top-ranked users under IID (Dir(10)) and non-IID (Dir(0.1)) settings in the presence of noisy users.

updates as a measure for distributional alignment, DuoShapley dynamically adjusts the weighting between individual and coalition-based contributions without requiring explicit knowledge of data heterogeneity. Beyond contribution estimation, DuoShapley supports a broad range of downstream use cases. It can guide user selection by identifying the most valuable participants under different data distributions, and detecting noisy or low-quality users whose contributions are negligible or detrimental to overall model performance. An overview of the framework is illustrated in Figure 1. Our key contributions are as follows:

- We provide an in-depth empirical analysis of two scalable coalition orders, Solo and LOO, revealing their complementary strengths and limitations under varying data heterogeneity in FL.

- We introduce DuoShapley, an efficient and adaptive SV approximation algorithm for estimating user contributions across FL scenarios with improved accuracy and scalability.

- We demonstrate the practical utility of DuoShapley through a user selection application, showing its ability to enhance robustness and efficiency in realistic FL settings.

## 2 Related Work

**Contribution Evaluation via Gradient Shapley Methods.** SV-based methods are widely used in FL to quantify user contributions (Wei et al., 2020; Song et al., 2019; Liu et al., 2022a). Gradient Shapley methods aim to eliminate the lengthy retraining of FL models by utilizing gradient updates of the users to approximate the FL sub-models for various users coalitions. Reference (Song et al., 2019) proposes two gradient Shapley methods: One-Round (OR) and Multi-Round (MR). OR calculates the SV once after the

training while MR calculates the SV in every FL round. Truncated Multi-Rounds Construction (TMR) (Wei et al., 2020) eliminates unnecessary FL sub-model reconstructions by adding a decay factor. TMC (Ghorbani & Zou, 2019) approximates SVs by sampling random permutations and truncating the evaluation process once marginal contributions become negligible. In GTG (Liu et al., 2022a), authors design a guided Monte Carlo sampling approach combined with truncation techniques to further improve the computation efficiency. Another line of research studies extreme levels of data heterogeneity, e.g., label imbalance, and propose variants of gradient Shapley methods for such scenarios (Huang et al., 2022; Yang et al., 2025). Existing approximations, including GTG, TMC, MR, and TMR, face significant computational challenges. Although they adopt different strategies to approximate user contributions, they all require evaluating a large number of coalitions, leading to high computational overhead. This limits their scalability and makes them impractical for large-scale FL deployments.

## 3  Problem Setup and Background

**Federated Learning (FL).** We consider a standard FL system with multiple users and one server. We denote the set of users by $\mathcal{K} = \{1, 2, \ldots, I\}$, where $I$ is the total number of users. Each user $i \in \mathcal{K}$ holds a local dataset $\mathcal{D}_i$ of size $n_i = |\mathcal{D}_i|$. Let $n = \sum_{i \in \mathcal{K}} n_i$ denote the total number of samples across all users. Each data point $(\boldsymbol{x}, y) \in \mathcal{D}_i$ consists of a feature vector $\boldsymbol{x}$ and a corresponding label $y$. Let $\mathcal{M} = \{1, 2, \ldots, C\}$ denote the set of class labels. The global model is parameterized by $\boldsymbol{w}$, and each user maintains a local model $\boldsymbol{w}_i$. The global objective is to minimize the empirical loss:

$$\min_{\boldsymbol{w}} \mathcal{L}(\boldsymbol{w}) = \sum_{i \in \mathcal{K}} \frac{n_i}{n} \mathcal{L}_i(\boldsymbol{w}), \tag{1}$$

where the local loss of user $i$ is defined as

$$\mathcal{L}_i(\boldsymbol{w}) = \frac{1}{n_i} \sum_{(\boldsymbol{x}, y) \in \mathcal{D}_i} \mathcal{L}(\boldsymbol{w}; \boldsymbol{x}, y). \tag{2}$$

The FL training process includes the following steps: (i) **Initialization:** The server initializes the global model parameters $\boldsymbol{w}$ and broadcasts them to all users. (ii) **Local Update and Model Aggregation:** In each communication round $t$, every user $i \in \mathcal{K} = \{1, 2, \ldots, I\}$ performs local training and sends the updated model parameters $\boldsymbol{w}_i^t$ to the server. The server then updates the global model by aggregating the received updates:

$$\boldsymbol{w}^{(t+1)} = \sum_{i=1}^{I} \frac{n_i}{n} \boldsymbol{w}_i^{(t)}. \tag{3}$$

Step (ii) is repeated until the global model converges.

**Shapley Value (SV) for User Valuation in FL.** SV (Shapley, 1971; Ghorbani & Zou, 2019) of user $i$ is given by

$$\phi_i(\mathcal{K}, \mathcal{V}) = \frac{1}{|\mathcal{K}|} \sum_{Q \subseteq \mathcal{K} \setminus \{i\}} \frac{\mathcal{V}(Q \cup \{i\}) - \mathcal{V}(Q)}{\binom{|\mathcal{K}|-1}{|Q|}}, \tag{4}$$

where $\phi_i$ is the SV for user $i$, and $Q$ denotes a subset of the participant set $\mathcal{K}$. The utility function $\mathcal{V}(\cdot)$ quantifies the performance or value of a given coalition of users. In this work, we define the utility function $\mathcal{V}(\cdot)$ as the validation accuracy measured on the server-side validation dataset, and use gradient-based Shapley methods for user contribution evaluation.

Computing the exact Shapley value in Equation (4) requires evaluating the utility over $|\mathcal{K}| \cdot 2^{|\mathcal{K}|-1}$ coalitions, which is computationally prohibitive even for moderate $|\mathcal{K}|$. Next, we define two computationally-efficient

coalition orders, Solo and Leave-One-Out (LOO). Each requires evaluating only $O(|\mathcal{K}|)$ coalitions and thus has linear complexity in the number of users.

For a utility function $\mathcal{V}(\cdot)$ and a user $i \in \mathcal{K}$, define the marginal contribution of user $i$ to a coalition $Q \subseteq \mathcal{K} \backslash \{i\}$ as

$$\Delta \mathcal{V}_i(Q) := \mathcal{V}(Q \cup \{i\}) - \mathcal{V}(Q). \tag{5}$$

**Definition 1** (Solo)**.** Solo evaluates user $i$ independently, and is defined as:

$$\phi_i^{\text{Solo}} := \Delta \mathcal{V}_i(\varnothing) = \mathcal{V}(\{i\}) - \mathcal{V}(\varnothing). \tag{6}$$

**Definition 2** (Leave-One-Out (LOO))**.** LOO evaluates the marginal contribution of user $i$ relative to the coalition of all other users, and is defined as:

$$\phi_i^{\text{LOO}} := \Delta \mathcal{V}_i(\mathcal{K} \backslash \{i\}) = \mathcal{V}(\mathcal{K}) - \mathcal{V}(\mathcal{K} \backslash \{i\}). \tag{7}$$

## 4 Proposed Method: DuoShapley

In this section, we analyze the behavior of Solo and LOO across different data distributions. Our analysis shows that Solo tends to perform well when user datasets are similar, while LOO becomes more effective in heterogeneous settings. Motivated by these observations, we propose DuoShapley, which adaptively combines Solo and LOO to balance efficiency and accuracy in estimating SVs across a wide range of FL scenarios.

### 4.1 Motivation

With the growing scale of FL deployments, efficiency is a key requirement for ensuring practical and scalable contribution evaluation. Instead of exploring complex coalition structures, we focus on two efficient endpoint coalition orders, Solo and LOO, as defined in Definitions 1 and 2. In our FL setting, Solo evaluates the validation accuracy of the model produced by user $i$ alone, while LOO compares the validation accuracy of the aggregated model using all users with that of the aggregated model obtained after excluding user $i$. Both scale linearly with the number of users, making them ideal for efficient SV estimation.

Importantly, Solo and LOO exhibit complementary strengths under different data distributions. Figure 2 analyzes the impact of adding and removing top-ranked users under IID and non-IID settings, in the presence of noisy users. For example, when the fraction is set to 0.3, the removal setting excludes the top 30% of users and retrains the model using the remaining participants. Conversely, the addition setting trains the model using only the top 30% of users. As shown in Figures 2a and 2b, Solo tends to perform well in IID settings, where user contributions are relatively uniform and individual evaluation suffices. In contrast, Figures 2c and 2d show that LOO is better suited for non-IID scenarios, where a user's value is revealed only in combination with others. Notably, LOO tends to capture top contributors, who are most critical to model performance, while Solo is more effective at identifying tail-end ones, highlighting their complementary strengths in user valuation.

Thus, each method alone fails to generalize well across all cases. Solo underestimates contributions in heterogeneous settings, and LOO lacks resolution in homogeneous ones. Motivated by this observation, we propose to dynamically combine Solo and LOO in our method, enabling more robust and distribution-aware user valuation in FL.

### 4.2 DuoShapley: Adaptive Weighting of Solo and LOO

To adaptively balance these two perspectives, we introduce a dynamic weighting scheme based on the cosine similarity between a user's update and the aggregated global update. Cosine similarity is a widely adopted metric in FL for measuring similarity based on direction, independent of magnitude. Recent studies in heterogeneous FL, robust aggregation, clustering, and personalization have demonstrated its effectiveness

in capturing similarity across diverse user distributions (Yan et al., 2024; Chen et al., 2024a; Mai et al., 2024; Li et al., 2024b; Sun et al., 2024). Building on these insights, we incorporate cosine similarity into our framework to capture meaningful relationships among user updates and to guide weighting of Solo and LOO.

Intuitively, when a user's update direction is well aligned with the aggregated global update, indicating consistent behavior with the majority, we place more weight on the Solo contribution. Conversely, when the cosine similarity is low, suggesting that the user may bring unique or complementary information, we shift more weight toward the LOO contribution. This mechanism allows our method to adjust smoothly across different data distributions without requiring explicit prior knowledge of the data heterogeneity.

We define the estimated SV of user $i$ as:

$$\phi_i^{(t)} = \alpha^{(t)} \cdot \phi_i^{\text{Solo},(t)} + (1 - \alpha^{(t)}) \cdot \phi_i^{\text{LOO},(t)}, \tag{8}$$

where $\phi_i^{\text{Solo},(t)}$ denotes the individual contribution of user $i$ at round $t$, and $\phi_i^{\text{LOO},(t)}$ represents their marginal contribution when removed from the full coalition, and $\alpha^{(t)} \in [0,1]$ balances the influence of each term.

We first explore using fixed values of $\alpha$, but no single value universally works well across all data heterogeneity levels. High $\alpha$ can overemphasize individual contributions in highly non-IID settings, leading to poor estimations. Conversely, low $\alpha$ can ignore useful individual signals in IID settings.

To address this, we design an adaptive weighting strategy that adjusts $\alpha$ based on the alignment between each user's local update and the aggregated global update. Specifically, we compute the cosine similarity $s_i^t$ between the local $\mathbf{\Delta}_i^{(t)}$ and the global $\mathbf{\Delta}^{(t)}$ for each user $i$ at round $t$:

$$s_i^{(t)} = \frac{\mathbf{\Delta}_i^{(t)} \cdot \mathbf{\Delta}^{(t)}}{\|\mathbf{\Delta}_i^{(t)}\|_2 \cdot \|\mathbf{\Delta}^{(t)}\|_2}, \tag{9}$$

where

$$\mathbf{\Delta}^{(t)} = \sum_{i=1}^{I} \frac{n_i}{\sum_{j=1}^{I} n_j} \mathbf{\Delta}_i^{(t)}, \quad \mathbf{\Delta}_i^{(t)} = \boldsymbol{w}_i^{(t)} - \boldsymbol{w}^{(t)}. \tag{10}$$

Cosine similarity $s_i^t$ ranges from $[-1, 1]$, where $+1$ indicates perfect alignment, 0 denotes orthogonality, and -1 implies complete opposition. In our setting, this cosine similarity serves as a proxy for the directional alignment between users' updates and the global update.

We then compute the $\alpha$ value for round $t$ as:

$$\alpha^{(t)} = \frac{1}{I} \sum_{i=1}^{I} \alpha_i^{(t)}, \quad \alpha_i^{(t)} := \text{clip}\left(\frac{s_i^{(t)}}{\tau}; 0, 1\right), \tag{11}$$

where clip$(\cdot)$ denotes the element-wise clipping operation defined as clip$(z; a, b) := \min(b, \max(a, z))$, which restricts the value of $z$ to lie within $[a, b]$. In Equation (11), $\tau$ acts as a tunable hyperparameter that controls how sharply the weighting transitions occur between Solo and LOO. Lower values of $\tau$ cause the weighting to shift rapidly toward Solo, while higher values result in a more gradual transition, giving more influence to LOO.

As shown in Equation (11), we apply clipped normalization to each user's similarity score $s_i^{(t)}$, mapping it to the range $[0, 1]$ to obtain $\alpha_i^{(t)}$. This transformation ensures a smooth transition between Solo and LOO weighting, where users with lower alignment receive weights closer to 0 (favoring LOO), and those with higher alignment receive weights closer to 1 (favoring Solo). Finally, we compute the global weighting factor $\alpha^{(t)}$ for round $t$ by averaging all users' normalized $\alpha_i^{(t)}$. This averaged $\alpha^{(t)}$ reflects the overall directional agreement among users in that round, guiding the balance between Solo and LOO in the SV estimation.

---

**Algorithm 1:** DuoShapley for User Valuation in FL

---

**1** **Input**: $T$: number of training rounds; $E$: number of local epochs; $\mathcal{K}$: set of users; $\mathcal{D}_i$: dataset of user $i$; $\mathcal{B}$: batch size; $n_i$: dataset size of user $i$; $\mathcal{V}_{val}(\cdot)$: utility function on validation dataset; $\tau$: normalization threshold; $\eta_i$: learning rate at user $i$.

**2** **Server executes**:

**3**    **for** each round $t = 0, \ldots, T-1$ **do**

**4**       **for** each user $i \in \mathcal{K}$ **in parallel do**

**5**          $\boldsymbol{w}_i^{(t)} \leftarrow \text{UserUpdate}(\boldsymbol{w}^{(t)}, i)$

**6**       $\boldsymbol{w}^{(t+1)} \leftarrow \sum_{i \in \mathcal{K}} \frac{n_i}{\sum_{j \in \mathcal{K}} n_j} \boldsymbol{w}_i^{(t)}$

**7**       $\{\phi_i^{(t)}\}_{i \in \mathcal{K}} \leftarrow \text{DuoShapley}\left(\{\boldsymbol{w}_i^{(t)}\}_{i \in \mathcal{K}}, \boldsymbol{w}^{(t)}, \boldsymbol{w}^{(t+1)}, \mathcal{V}\right)$

**8** **function** $\text{UserUpdate}(\boldsymbol{w}^{(t)}, i)$:

**9**    **for** each local epoch $e = 1, \ldots, E$ **do**

**10**       **for** each batch $\mathcal{D}_i^{(\mathcal{B})} \subseteq \mathcal{D}_i$ of size $\mathcal{B}$ **do**

**11**          $\boldsymbol{w}_i^{(t)} \leftarrow \boldsymbol{w}^{(t)} - \eta_i \nabla \mathcal{L}_i(\mathcal{D}_i^{(\mathcal{B})}, \boldsymbol{w}^{(t)})$

**12**       **end for**

**13**    **return** $\boldsymbol{w}_i^{(t)}$ to server

**14** **function** $\text{DuoShapley}(\{\boldsymbol{w}_i^{(t)}\}, \boldsymbol{w}^{(t)}, \boldsymbol{w}^{(t+1)}, \mathcal{V})$:

**15**    **for** each user $i \in \mathcal{K}$ **do**

**16**       $\phi_i^{\text{Solo},(t)} \leftarrow \mathcal{V}(\boldsymbol{w}_i^{(t)}) - \mathcal{V}(\boldsymbol{w}^{(t)})$

**17**       $\boldsymbol{w}_{-i}^{(t+1)} \leftarrow$ Aggregated model without user $i$

**18**       $\phi_i^{\text{LOO},(t)} \leftarrow \mathcal{V}(\boldsymbol{w}^{(t+1)}) - \mathcal{V}(\boldsymbol{w}_{-i}^{(t+1)})$

**19**       $\boldsymbol{\Delta}_i^{(t)} \leftarrow \boldsymbol{w}_i^{(t)} - \boldsymbol{w}^{(t)}$ for each $i \in \mathcal{K}$

**20**       $\boldsymbol{\Delta}^{(t)} \leftarrow \sum_{i \in \mathcal{K}} \frac{n_i}{\sum_{j \in \mathcal{K}} n_j} \boldsymbol{\Delta}_i^{(t)}$

**21**       $s_i^{(t)} \leftarrow \text{CosSim}(\boldsymbol{\Delta}_i^{(t)}, \boldsymbol{\Delta}^{(t)})$ for each $i \in \mathcal{K}$

**22**       $\alpha_i^{(t)} \leftarrow \text{clip}(s_i^{(t)}/\tau; \ 0, \ 1)$ for each $i \in \mathcal{K}$

**23**       $\alpha^{(t)} \leftarrow \frac{1}{|\mathcal{K}|} \sum_{i \in \mathcal{K}} \alpha_i^{(t)}$

**24**    **for** each user $i \in \mathcal{K}$ **do**

**25**       $\phi_i^{(t)} \leftarrow \alpha^{(t)} \cdot \phi_i^{\text{Solo},(t)} + (1 - \alpha^{(t)}) \cdot \phi_i^{\text{LOO},(t)}$

**26**    **return** $\{\phi_i^{(t)}\}_{i \in \mathcal{K}}$

---

As shown in Algorithm 1, the server first collects model updates from all users. It then computes the Solo and LOO contributions for each user based on their uploaded updates. To balance Solo and LOO, the server calculates a weight $\alpha_i^{(t)}$ for each user by applying clipped normalization to their cosine similarity score $s_i^{(t)}$, as defined in Equations (9) and (11). Finally, the overall Shapley-based contribution $\phi_i^{(t)}$ for each user is computed using the weighted combination in Equation (8).

**Cumulative Shapley Score.** To maintain a stable and historical view of user contributions, we compute the Exponential Moving Average (EMA) of estimated SVs across rounds (Liao et al., 2025; Zhou et al., 2025; Kim et al., 2024; Wang et al., 2023; Nagalapatti & Narayanam, 2021). Let $R_i^{(t)}$ denote the EMA for user $i$ at round $t$. Then we have:

$$R_i^{(t)} = \beta \cdot R_i^{(t-1)} + (1 - \beta) \cdot \phi_i^{(t)}, \tag{12}$$

where $\beta \in [0, 1]$ is a smoothing factor. A higher $\beta$ gives more weight to past estimates, leading to more stable accumulation over time.

## 5 Theoretical Analysis

This section provides a theoretical analysis of DuoShapley. We first show that DuoShapley has linear complexity. We then derive an error decomposition with respect to the exact Shapley value. Finally, under a

monotone submodular utility model, we show that Solo and LOO form two principled extreme coalition orders that bound the exact Shapley value and motivate its approximation through an interpolation coefficient.

### 5.1 Preliminaries

We extend the notation in Definitions 1 and 2 to a fixed FL round $t$. Let $\mathcal{K}$ denote the user set with $|\mathcal{K}| = I$, and let $\mathcal{V}^{(t)} : 2^{\mathcal{K}} \to \mathbb{R}$ denote the coalition utility evaluated on the server side validation dataset at round $t$. For any user $i \in \mathcal{K}$ and coalition $Q \subseteq \mathcal{K} \setminus \{i\}$, the round-wise marginal contribution is

$$\Delta \mathcal{V}_i^{(t)}(Q) := \mathcal{V}^{(t)}(Q \cup \{i\}) - \mathcal{V}^{(t)}(Q). \tag{13}$$

The exact Shapley value of user $i$ at round $t$ is

$$\phi_i^{\star,(t)} := \sum_{Q \subseteq \mathcal{K} \setminus \{i\}} \frac{|Q|! \, (I - |Q| - 1)!}{I!} \, \Delta \mathcal{V}_i^{(t)}(Q), \tag{14}$$

i.e., the weighted average of user $i$'s marginal contributions over all coalitions under the round-wise utility $\mathcal{V}^{(t)}$.

Accordingly, the two endpoint coalition orders are written as

$$\phi_i^{\mathrm{Solo},(t)} := \Delta \mathcal{V}_i^{(t)}(\varnothing) = \mathcal{V}^{(t)}(\{i\}) - \mathcal{V}^{(t)}(\varnothing), \tag{15}$$

$$\phi_i^{\mathrm{LOO},(t)} := \Delta \mathcal{V}_i^{(t)}(\mathcal{K} \setminus \{i\}) = \mathcal{V}^{(t)}(\mathcal{K}) - \mathcal{V}^{(t)}(\mathcal{K} \setminus \{i\}), \tag{16}$$

DuoShapley estimates the round-wise contribution of user $i$ by interpolating between these two endpoints:

$$\phi_i^{(t)} = \alpha^{(t)} \phi_i^{\mathrm{Solo},(t)} + (1 - \alpha^{(t)}) \phi_i^{\mathrm{LOO},(t)}, \qquad \alpha^{(t)} \in [0, 1]. \tag{17}$$

### 5.2 Linear Complexity

We first formalize the computational advantage of DuoShapley. Since DuoShapley only evaluates the two endpoint coalition orders, Solo and LOO, its cost grows linearly with the number of users.

**Theorem 1** (Linear complexity). *For a fixed round $t$, let $C_V$ denote the cost of one utility evaluation on the validation set, and let $p$ denote the model dimension. DuoShapley requires $O(I)$ utility evaluations and $O(Ip)$ additional arithmetic operations. Hence its per-round complexity is*

$$O(IC_V + Ip), \tag{18}$$

*which is linear in the number of users.*

*Proof.* For each user $i$, DuoShapley computes one Solo utility and one LOO utility, so the number of utility evaluations is $O(I)$. Since each utility evaluation costs $C_V$, this contributes $O(IC_V)$. DuoShapley also computes cosine similarity weight per user and aggregates them into a shared round level coefficient. Each similarity computation costs $O(p)$, so this contributes $O(Ip)$ overall. Combining the two terms gives $O(IC_V + Ip)$, which is linear in the number of users. □

### 5.3 Error Analysis

Efficiency alone is not sufficient; we also need to understand what approximation error DuoShapley makes compared to the exact Shapley value. Theorem 2 gives a round-wise error decomposition that isolates two sources of error in our estimator and provides bounds, without any assumptions.

**Theorem 2** (Error decomposition). *Fix a round $t$ and user $i$. Let $\alpha_i^{\star,(t)} \in [0,1]$ denote the optimal coefficient, and define the residual*

$$r_i^{(t)} := \phi_i^{\star,(t)} - \left(\alpha_i^{\star,(t)}\phi_i^{\text{Solo},(t)} + (1-\alpha_i^{\star,(t)})\phi_i^{\text{LOO},(t)}\right). \tag{19}$$

*Then the DuoShapley estimation error satisfies*

$$\left|\phi_i^{(t)} - \phi_i^{\star,(t)}\right| \leq \left|r_i^{(t)}\right| + \left|\alpha^{(t)} - \alpha_i^{\star,(t)}\right|\left|\phi_i^{\text{Solo},(t)} - \phi_i^{\text{LOO},(t)}\right|. \tag{20}$$

*Proof.* See Appendix C.1. □

Theorem 2 shows that DuoShapley has two distinct error components. The first is a *weighting error*, $\left|\alpha^{(t)} - \alpha_i^{\star,(t)}\right|\left|\phi_i^{\text{Solo},(t)} - \phi_i^{\text{LOO},(t)}\right|$, which reflects how far the computed coefficient $\alpha^{(t)}$ is from the optimal interpolation coefficient $\alpha_i^{\star,(t)}$ between Solo and LOO. The second is a *residual error*, $|r_i^{(t)}|$, which captures the approximation gap introduced by skipping intermediate coalition orders. Thus, the weighting error comes from estimating the interpolation coefficient, while the residual error comes from restricting to the two endpoint coalition orders, Solo and LOO.

**Special Case: Monotone Submodular Utility.** Next, we ask when focusing only on Solo and LOO is well justified. To facilitate analysis, we make the mild assumption of monotone submodular utility model, where marginal contributions decrease with coalition size. We show that, in this case, Solo and LOO are the upper and lower bounds on marginal contributions.

**Assumption 1** (Monotone submodular utility). *For a fixed round $t$, the utility function $\mathcal{V}^{(t)}$ is:*

- ***monotone**: $\mathcal{V}^{(t)}(A) \leq \mathcal{V}^{(t)}(B)$ whenever $A \subseteq B$;*

- ***submodular**: for all $A \subseteq B \subseteq \mathcal{K} \setminus \{i\}$,*

$$\Delta\mathcal{V}_i^{(t)}(A) \geq \Delta\mathcal{V}_i^{(t)}(B). \tag{21}$$

**Theorem 3** (Bounds on coalition marginals). *Under Assumption 1, for every round $t$, user $i$, and coalition $Q \subseteq \mathcal{K} \setminus \{i\}$,*

$$\phi_i^{\text{LOO},(t)} \leq \Delta\mathcal{V}_i^{(t)}(Q) \leq \phi_i^{\text{Solo},(t)}. \tag{22}$$

*Consequently,*

$$\phi_i^{\text{LOO},(t)} \leq \phi_i^{\star,(t)} \leq \phi_i^{\text{Solo},(t)}. \tag{23}$$

*Proof.* See Appendix C.2 □

Theorem 3 demonstrates our intuition of why the DuoShapley design focuses on the two endpoint coalition orders, Solo and LOO. Under monotone submodularity, a user's marginal contribution decreases as the coalition grows, so every intermediate coalition marginal lies between these two extremes. Since the exact Shapley value is a weighted average of all such marginal contributions, it must also lie between Solo and LOO.

**Corollary 1** (Exact interpolation between Solo and LOO). *Under Assumption 1, if*

$$\phi_i^{\text{Solo},(t)} \neq \phi_i^{\text{LOO},(t)}, \tag{24}$$

*then there exists an interpolation coefficient*

$$\alpha_i^{\star,(t)} = \frac{\phi_i^{\star,(t)} - \phi_i^{\text{LOO},(t)}}{\phi_i^{\text{Solo},(t)} - \phi_i^{\text{LOO},(t)}} \in [0,1] \tag{25}$$

*such that*

$$\phi_i^{\star,(t)} = \alpha_i^{\star,(t)} \phi_i^{\text{Solo},(t)} + (1 - \alpha_i^{\star,(t)}) \phi_i^{\text{LOO},(t)}. \tag{26}$$

*Hence the residual in Theorem 2 is exactly zero:*

$$r_i^{(t)} = 0. \tag{27}$$

*Proof.* See Appendix C.3 $\qquad\qquad\qquad\qquad\qquad\qquad\qquad\qquad\qquad\qquad\qquad\qquad\square$

Corollary 1 shows that, under monotone submodularity, the exact Shapley value can be written exactly as a weighted average of Solo and LOO. Thus, ignoring intermediate coalition orders does not introduce a separate representation error; the only remaining approximation error comes from how well the adaptive weight $\alpha^{(t)}$ matches the optimal coefficient $\alpha_i^{\star,(t)}$. In the main paper, we compute $\alpha^{(t)}$ using the cosine similarity based method in Equations (9) and (11), which uses the alignment between local and global updates for balancing Solo and LOO. Stronger alignment favors Solo, while weaker alignment points to stronger complementarity and therefore favors LOO.

**Scope of assumptions.** The monotone submodular analysis above justifies the design of DuoShapley from a theoretical perspective. In practical federated learning, validation accuracy may not be strictly monotone or submodular due to optimization noise, label noise, or noisy participants. Theorem 2 holds without additional assumptions on the utility function and therefore applies generally. Theorem 3 shows that, under monotone submodularity, Solo and LOO bound the intermediate marginal contributions. Corollary 1 further shows that the exact Shapley value can be written exactly as an interpolation between Solo and LOO.

## 6 Experiments

We comprehensively evaluate the effectiveness of DuoShapley across varying levels of data heterogeneity and two user scales: 10, and 50 users. Each setting is designed to examine different aspects of SV approximation. These experiments enable a systematic evaluation of DuoShapley across diverse settings, ranging from small-scale scenarios where SV approximations are feasible to large-scale systems where efficiency and adaptability are essential.

### 6.1 Evaluation Setup

**Datasets and Models.** We use two common benchmark datasets, (i) CIFAR-10 (Krizhevsky et al., 2009) consisting of colored images of 10 classes, with 50,000 samples for training and 10,000 for testing, and (ii) Fashion-MNIST (F-MNIST) (Xiao et al., 2017) consisting of fashion images, with 60,000 samples for training and 10,000 for testing. For both CIFAR-10 and F-MNIST, we randomly split 20% of testing samples as validation dataset. We utilize a commonly employed CNN model (Albawi et al., 2017) for the CIFAR-10 and F-MNIST datasets. Additional experiments on ImageNet-100 are provided in Appendix B.

**Data Heterogeneity Scenarios.** Both CIFAR-10 and F-MNIST datasets are uniformly distributed across all 10 class labels. To simulate data heterogeneity, we adopt the widely used practical heterogeneous setting based on the Dirichlet distribution (Zhang et al., 2023), denoted as $Dir(\gamma)$. We consider four levels of heterogeneity with $\gamma \in \{10, 1, 0.5, 0.1\}$ across all datasets. The parameter $\gamma$ controls the level of heterogeneity: higher values lead to more IID-like data partitions across users, while lower values introduce greater heterogeneity. Specifically, $Dir(10)$ reflects the extreme IID scenario, where user data distributions are nearly identical, and $Dir(0.1)$ represents the most heterogeneous case, where user distributions differ substantially.

**FL Setup.** We train for 100 rounds in the 10-user setting, and for 200 rounds in the 50-user setting for both F-MNIST and CIFAR-10 datasets. The batch size is 64; the learning rate is 0.05 for all baselines in both datasets. Each selected user performs one local epoch per communication round. All results are averaged over three runs.

**Baselines.** MR (Song et al., 2019); TMR (Wei et al., 2020); TMC (Ghorbani & Zou, 2019); GTG (Liu et al., 2022a); LOO (Ghorbani & Zou, 2019).

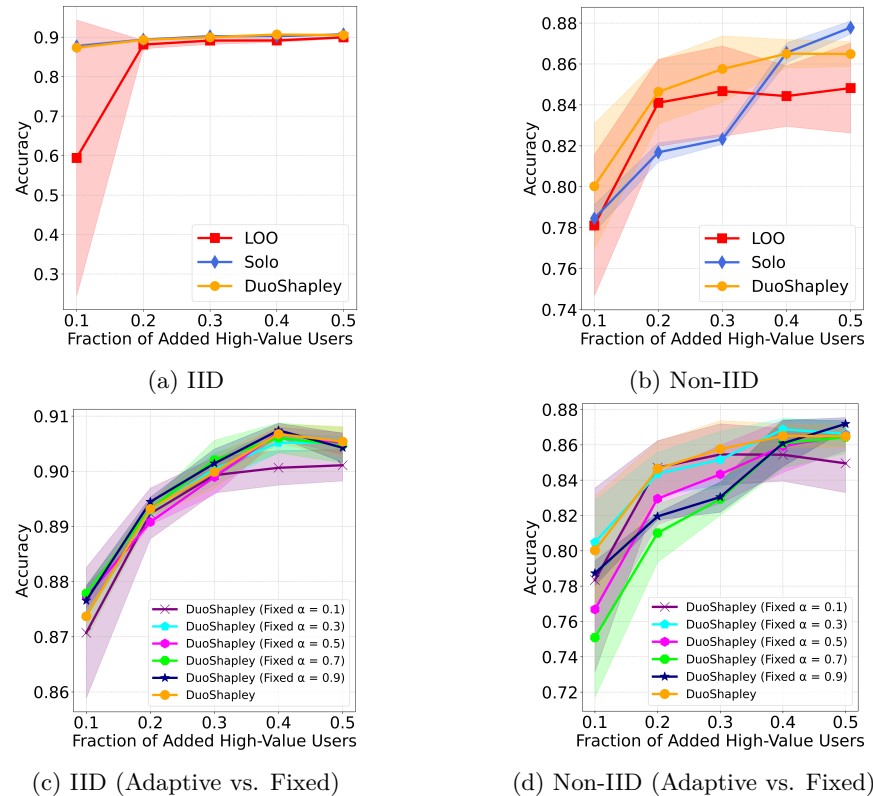

(a) IID

(b) Non-IID

(c) IID (Adaptive vs. Fixed)

(d) Non-IID (Adaptive vs. Fixed)

Figure 3: **User addition analysis under IID and non-IID settings.** We evaluate the impact of adding top-ranked users (from 10% to 50%) in the 50-user setting with noisy users. (a) and (b) show results under IID (Dir(10)) and non-IID (Dir(0.1)) distributions, respectively, comparing Solo, LOO, and DuoShapley. (c) and (d) compare DuoShapley with the adaptive $\alpha$ against the fixed $\alpha$ variants under IID and non-IID settings, respectively.

Table 1: **Per-round runtime (in seconds) of each method for varying numbers of users.** All methods use FedAvg-style aggregation, with cost dominated by model utility evaluation.

| Method | 5 Users | 10 Users | 50 Users | 100 Users |
|---|---|---|---|---|
| MR | 8.70 | 224.18 | ✗ | ✗ |
| TMR | 8.34 | 222.40 | ✗ | ✗ |
| GTG | 8.58 | 153.20 | 10101.86 | ✗ |
| TMC | 8.73 | 118.17 | 6017.78 | ✗ |
| LOO | 1.91 | 3.62 | 13.71 | 25.69 |
| Solo | 1.93 | 3.51 | 13.21 | 24.25 |
| DuoShapley | 3.24 | 6.61 | 23.93 | 47.41 |

**10 User Setting.** In this setting, we assess the practical utility of each SV approximation by removing users with the highest estimated SVs (top 10%, 30%) and measuring the resulting drop in model accuracy. This tests how effectively each method identifies the most important users. We also measure the runtime of each method to highlight differences in computational efficiency.

**50 User Setting.** This setting simulates a practical FL environment, where the slow runtime of baseline methods limits their scalability to larger deployments. We therefore focus on the most efficient approaches: Solo, LOO, and the proposed DuoShapley, to evaluate scalability and robustness. In this setup, we include 50 benign users and 25 additional noisy users (approximately 33% of the total 75 participants) to simulate

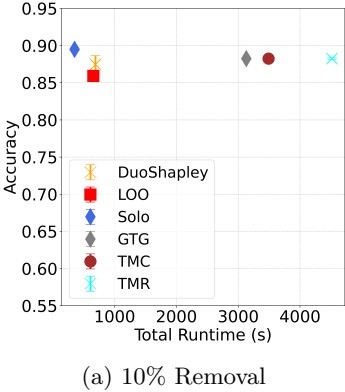
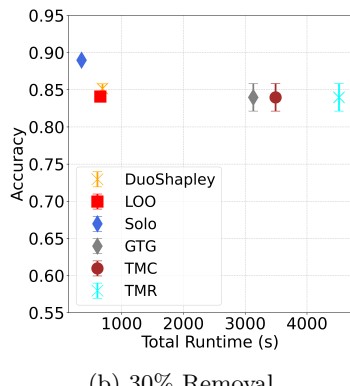

(a) 10% Removal         (b) 30% Removal

Figure 4: **Trade-off between accuracy and runtime under user removal in non-IID setting.** We evaluate all baselines and DuoShapley in a 10-user FL setting with non-IID (Dir(0.1)). The x-axis denotes total runtime, and the y-axis reflects test accuracy. (a) and (b) correspond to top-ranked user removal rates of 10% and 30%.

practical conditions involving low-quality and noisy users. We introduce noisy users to reflect real-world scenarios where some participants may produce unreliable updates caused by label noise, data quality issues, or inconsistent local optimization.

## 6.2 Evaluation of Efficiency and Scalability

Table 1 reports the average runtime (in seconds) per round for each SV approximation method across four user scales. For small-scale settings (5 and 10 users), all methods are tractable, though the baselines (MR, TMR, GTG, TMC) are significantly slower than LOO, Solo, and DuoShapley. As the number of users increases, the inefficiency of the baseline methods becomes more apparent: for 50 users, TMC and GTG require over 6,000 and 10,000 seconds per round, respectively, while MR and TMR become even more impractical (due to exponential complexity).

In contrast, Solo and LOO maintain low runtimes and scale efficiently, showing linear growth with respect to the number of users. Since DuoShapley is a linear combination of Solo and LOO, it inherits their efficiency and benefits in practice. For example, DuoShapley is 200× faster than GTG and TMC at 50 users. These results highlight the practical advantage of DuoShapley as a scalable and efficient approximation suitable for real-world FL deployments.

## 6.3 Evaluation of SV Approximation Accuracy

**10-User Evaluation.** This setting simulates how well each method identifies the most critical contributors. Figures 4a and 4b show that all methods behave similarly in terms of accuracy when a small fraction of users is removed (top 10% and 30%) despite GTG, TMR, and TMC taking more than 3× longer. When runtime is taken into account, our method, DuoShapley, along with other efficient baselines like LOO and Solo, clearly stands out by providing a better trade-off between accuracy and runtime efficiency.

**50-User Evaluation.** As shown in Table 1, computing SVs using GTG and TMC becomes over 200× slower compared to LOO and Solo for 50 users, making these baseline methods impractical at scale. Due to their computational inefficiency, we narrow our focus to the most scalable methods: Solo, LOO, and DuoShapley.

Figures 3a and 3b compare Solo, LOO, and DuoShapley under IID and non-IID conditions, respectively. To assess their effectiveness, we simulate the process of incrementally adding users with the highest estimated SVs. Solo and LOO excel in different regimes: Solo performs best in IID settings, while LOO is more effective under non-IID distributions. DuoShapley, by adaptively leveraging both, consistently achieves superior or comparable performance across both settings. Specifically, for the top 10%, 20%, and 30% most valuable user additions, DuoShapley outperforms both Solo and LOO under non-IID conditions.

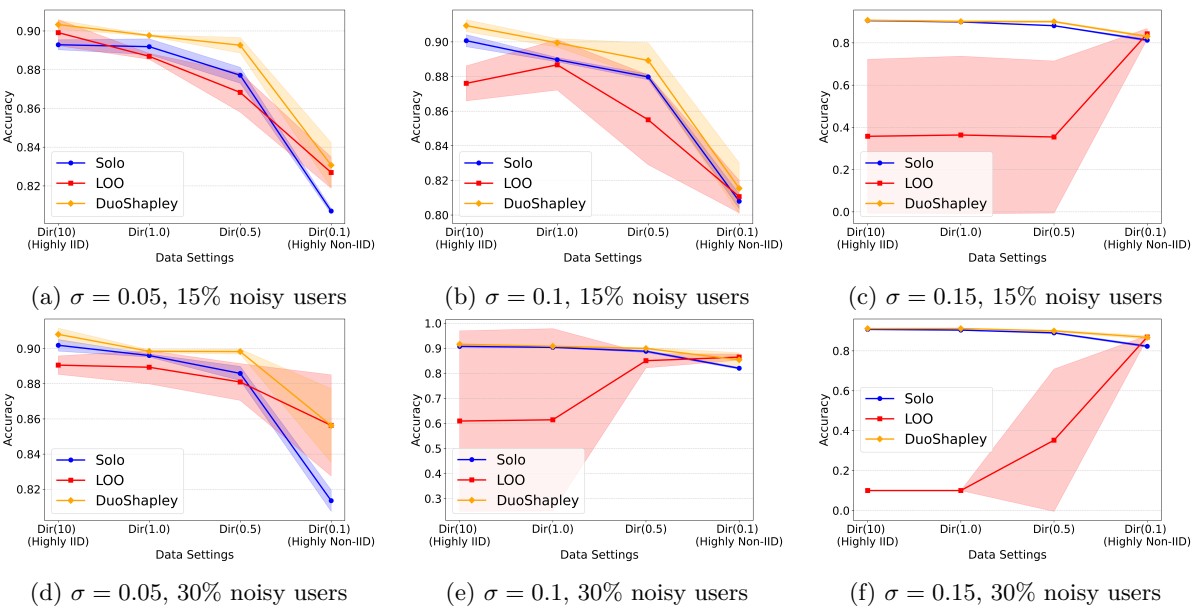

Figure 5: **Robust user selection in the presence of noisy users.** Test accuracy comparison of DuoShapley across different data distributions and noise levels. Experiments are conducted with 15% and 30% noisy users, injected with noise sampled from a Gaussian distribution with zero-mean and standard deviation $\sigma$ (i.e., noise level), where $\sigma \in \{0.05, 0.1, 0.15\}$. In each round, the server selects the top 20% of users based on the ranking of their estimated contributions. Every 5 rounds, the server queries updates from all users to update their contribution estimates.

Beyond the addition of top 40%, Solo starts to slightly outperform DuoShapley. This happens because Solo evaluates each user's contribution independently, allowing it to better recognize the value of users with less obvious impact, especially toward the tail. In contrast, LOO excels at identifying users whose contributions significantly benefit the overall coalition, which leads to high-value users being ranked near the top. However, it tends to overlook those users at the tail-end whose contributions are smaller or harder to spot within the coalition. Overall, DuoShapley is consistently competitive across different data heterogeneity levels. It strikes a good balance between Solo and LOO and is particularly effective at identifying the most valuable users, who play a key role in practical FL deployments where only a small subset of strong contributors can be selected from the entire user pool.

Moreover, as formulated in Equation (8), DuoShapley adaptively combines Solo and LOO using a balancing parameter $\alpha$. The comparisons in Figures 3c and 3d further demonstrate the advantage of using an adaptive $\alpha$. The adaptive $\alpha$ strategy achieves performance closely aligned with the best fixed $\alpha$ across both IID and non-IID settings, without requiring manual selection of a fixed $\alpha$.

# 7 Practical Applications of DuoShapley

In real-world FL deployments, users often vary in data quality, availability, and reliability. Involving all users in every round is typically impractical due to resource constraints, communication overhead, and the existence of noisy or low-quality participants (Zuo et al., 2025; Min et al., 2026). This motivates the need for a principled user selection mechanism that prioritizes those who contribute most to model improvement.

We design a selection mechanism based on the cumulative Shapley score, as defined in Equation (12). In each round, the server selects the top $\rho$ fraction of users based on their cumulative Shapley scores, which are updated every $\nu$ rounds using model updates collected from all users. In our experiments, we set $\rho = 0.2$ and $\nu = 5$. To evaluate the selection mechanism, we simulate noisy user scenarios with 15% and 30% noisy users and noise levels of 0.05, 0.1, and 0.15. Noisy users return random updates drawn from a zero-mean

Gaussian distribution, with the standard deviation $\sigma$ corresponding to the noise level. These settings allow us to compare Solo, LOO, and DuoShapley in selecting high-quality users under challenging conditions.

As shown in Figure 5, Solo performs well in IID and moderately non-IID settings (Dir(10), Dir(1), Dir(0.5)). In contrast, LOO excels in highly non-IID settings (Dir(0.1)), as demonstrated in Figures 5a and 5d, where user data distributions are significantly heterogeneous. However, the performance of LOO is sensitive to the noise level and the proportion of noisy users. As noise increases, Figures 5b and 5e show that LOO's coalition-based valuation becomes more affected by unreliable participants, degrading its estimation accuracy. Conversely, Solo's focus on individual contribution makes it more resilient to noise, exhibiting more stable performance under higher noise conditions, as shown in Figures 5c and 5f. DuoShapley effectively inherits the strengths of both Solo and LOO, aligning closely with Solo in IID settings and more with LOO in non-IID scenarios, achieving robust performance throughout. Overall, Figure 5 demonstrates the consistent performance of DuoShapley across all settings, highlighting its practical value for real-world FL deployments.

## 8 Conclusion

In this work, we present DuoShapley, an efficient and adaptive algorithm for user valuation in FL. By leveraging the complementary strengths of Solo and LOO contributions, DuoShapley dynamically adjusts their influences through a cosine-based weighting strategy that reflects the alignment between local and global model updates, enabling effective SV approximation across both IID and heterogeneous data distributions. DuoShapley scales linearly with the number of users and achieves over $200\times$ speedup in per-round runtime compared to existing methods such as GTG and TMC, making it highly suitable for practical and large-scale FL deployments. To further highlight its applicability in practice, we apply DuoShapley to a user selection application, demonstrating its effectiveness in identifying the most valuable participants and enhancing both robustness and efficiency. Overall, DuoShapley strikes a practical balance between computational efficiency and valuation accuracy, and offers a scalable solution for contribution evaluation in real-world FL systems.

### Acknowledgments

This work was partially supported by National Science Foundation (NSF) Awards 1956393 and 1900654 to UC Irvine, and by a generous gift from the Samueli Foundation to the E+S Institute on AI (a.k.a. ProperAI).

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

# A    Additional Experiments on F-MNIST and CIFAR-10

This appendix extends the main body of our paper, providing supplementary materials to enhance the understanding of DuoShapley.

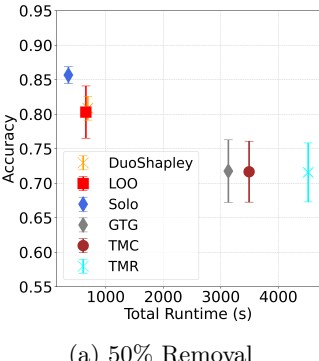

(a) 50% Removal

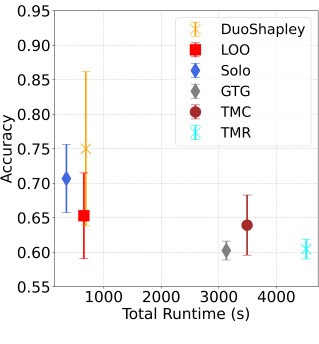

(b) 70% Removal

Figure 6: **Trade-off between accuracy and runtime under user removal in non-IID setting.** We evaluate all baselines and DuoShapley in a 10-user FL setting without noisy users, under a non-IID distribution (Dir(0.1)). The x-axis denotes total runtime, and the y-axis reflects test accuracy. (a) and (b) correspond to top-ranked user removal rates of 50%, and 70%, respectively, with users removed according to their estimated SV rankings.

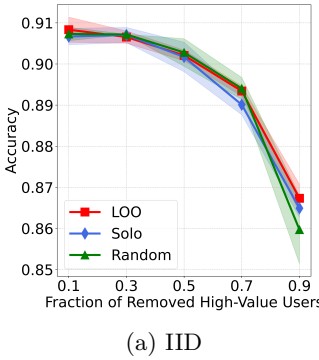

(a) IID

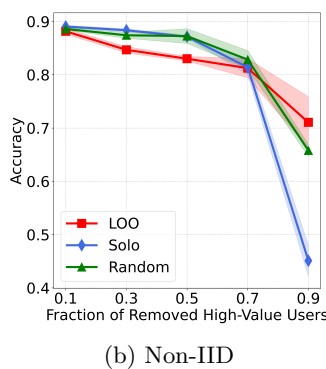

(b) Non-IID

Figure 7: **User removal under IID and non-IID settings.** We evaluate the effectiveness of efficient coalition orders, Solo, and the LOO, for estimating user contributions, with Random included as a baseline. The analysis is conducted by removing top-ranked users under IID (Dir(10)) and non-IID (Dir(0.1)) settings without the presence of noisy users.

## A.1    Additional Results on F-MNIST

**10 User Setting.** We evaluate the utility of each method by removing users with the highest estimated SVs and observing the resulting drop in model accuracy. As the removal rate increases to 50% and 70% (Figures 6a and 6b), baseline methods such as GTG, TMR, and TMC lead to larger accuracy drops, suggesting stronger alignment with key contributors. However, these higher removal rates of 50% and 70% mostly involve users ranked in the middle or near the end of the contribution list. In practice, large-scale FL systems rarely select such a large fraction of participants per round. Instead, they prioritize identifying the top contributors, such as the top 10–30%, who drive most of the model improvement. Our primary focus is on evaluating how well each method identifies and ranks these most valuable users.

**50 User Setting.** Figure 7 examines the impact of removing top-ranked users in both IID and non-IID settings, without any noisy participants. In the IID case (Figure 7a), all methods perform similarly because,

with homogeneous data and no noisy users, participants contribute equally. As a result, removing any of them leads to similar effects. In contrast, Figure 7b shows that LOO performs better in non-IID settings, where a user's value emerges only through interaction with others. Notably, LOO is more effective at identifying top contributors who are most critical to overall model performance.

## A.2 Additional Results on CIFAR-10

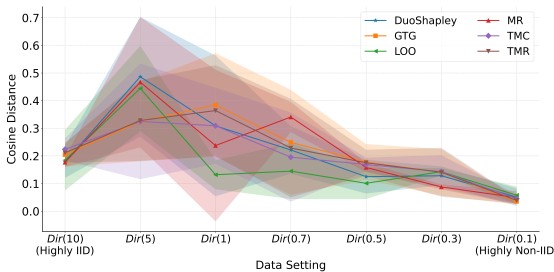

Figure 8: **Evaluation of SV approximation accuracy on CIFAR-10.** Approximation accuracy is assessed across seven levels of data heterogeneity using the Cosine Distance metric. Lower values indicate more accurate approximations.

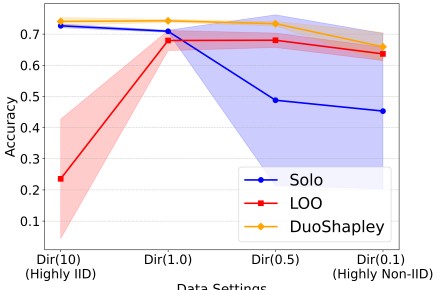

Figure 9: **Robust user selection in the presence of noisy users on CIFAR-10.** Test accuracy comparison of DuoShapley across different data distributions. Experiments are conducted with 30% noisy users, injected with random noise level 0.06. In each round, the server selects the top 20% of users based on the ranking of their estimated contributions. Every 5 rounds, the server queries updates from all users to update their contribution estimates.

**5 User Setting.** Given the small number of users, exact Shapley values (ExactSV) and all approximation methods can be computed efficiently. We use ExactSV as the ground truth to evaluate the accuracy of each approximation. To assess the effectiveness of evaluated mechanisms, we report the Cosine Distance as the accuracy metric for estimating the distance between ExactSV and approximations. Prior to distance calculation, we apply min-max normalization to all Shapley value vectors to ensure fair comparison across metrics and eliminate scale-related bias. As shown in Figure 8, all methods yield similar results in this small-scale setting, with error metrics tightly clustered and performance differences marginal. This outcome is expected, as the limited number of users reduces combinatorial complexity and flattens the variation among approximations.

**User Selection with CIFAR-10** Figure 9 presents results under the CIFAR-10 setting with 50 benign users, and 22 noisy participants (noise level 0.06). Solo performs well in IID settings but its effectiveness diminishes as data heterogeneity increases. In contrast, LOO excels in highly non-IID scenarios, where collaborative effects play a larger role, but performs worse under IID settings due to its reliance on coalition-based evaluation. DuoShapley effectively combines the strengths of both, aligning with Solo in IID and LOO in non-IID settings.

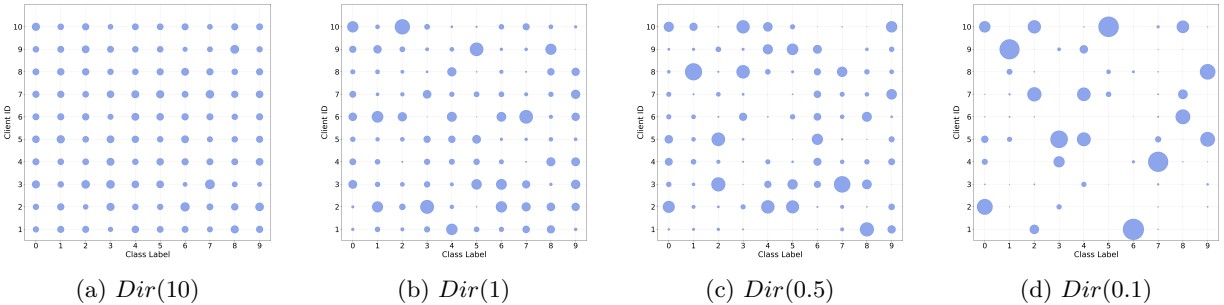

Figure 10: **Data distribution across 10 users in four scenarios.** Circle size indicates sample count, with heterogeneity controlled by $\gamma$ in $Dir(\gamma)$. Higher $\gamma$ means lower heterogeneity. (a) is the most IID setting, with highly identical class distributions across users. (b) and (c) show moderate heterogeneity. (d) is the most non-IID.

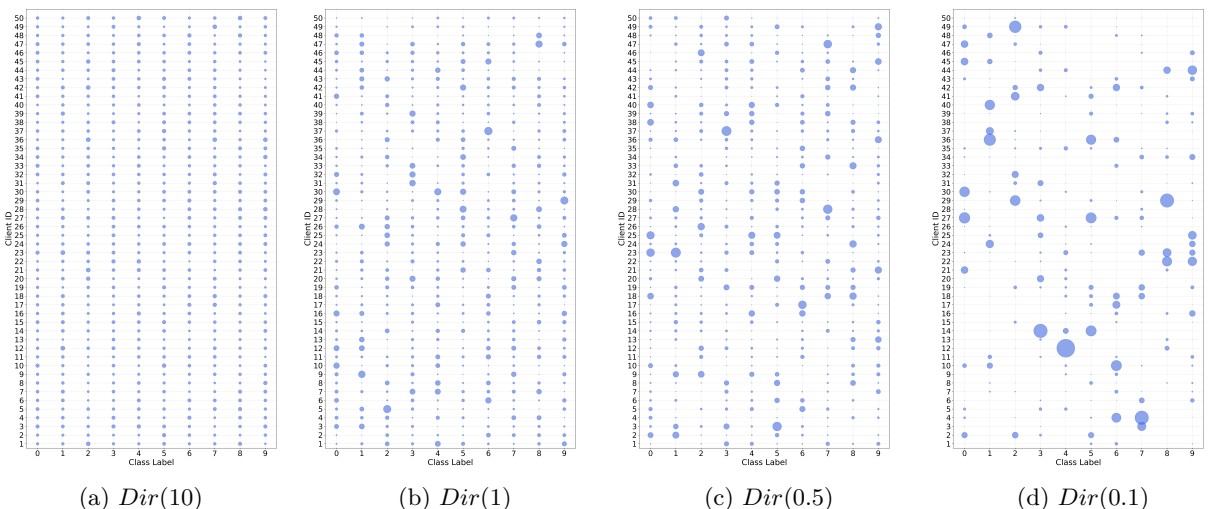

Figure 11: **Data distribution across 50 users in four scenarios.** Circle size indicates sample count, with heterogeneity controlled by $\gamma$ in $Dir(\gamma)$. Higher $\gamma$ means lower heterogeneity. (a) is the most IID setting, with highly identical class distributions across users. (b) and (c) show moderate heterogeneity. (d) is the most non-IID.

### A.3 Data Heterogeneity Scenarios

Figures 10 and 11 visualize the distribution of data across users for four levels of heterogeneity, from highly IID (Dir(10)) to highly non-IID (Dir(0.1)), under 10 and 50 users settings, respectively.

### A.4 Hyperparameter Settings

In our experiments, we set $\beta = 0.8$ for all baseline methods in user removal and addition tasks, and $\beta = 0.9$ for the user selection application. For DuoShapley, we use $\tau = 1.0$ in the 10-user setting. In the 50-user setting, we set $\tau = 0.3$ for user addition experiments and $\tau = 1.0$ for user selection. For all truncation-based baselines, we use a round truncation threshold of 0.01. All experiments are implemented in PyTorch and conducted on two NVIDIA RTX A5000 GPUs and two Intel Xeon Silver 4316 CPUs.

## B  Additional Experiments on ImageNet-100

To evaluate DuoShapley in a larger setting, we provide additional experiments on the ImageNet-100 dataset (ImageNet-100, 2024; Tian et al., 2020), which is a 100-class subset of ImageNet (Deng et al., 2009). Compared with CIFAR-10 and Fashion-MNIST used in our main paper, ImageNet-100 contains 100 classes, higher-resolution images, and more data samples, making it a larger-scale and more practical benchmark. We also evaluate a larger-scale setting with 100 users and a pretrained model ResNet-18 (He et al., 2016) under classifier fine-tuning, extending beyond the scale of the existing experiments in the paper.

Table 2: **Per-round runtime (in seconds) of each method for varying numbers of users on ImageNet-100.** All methods use FedAvg-style aggregation.

| Method | 5 Users | 10 Users | 50 Users | 100 Users | 1000 Users |
|---|---|---|---|---|---|
| MR | 64.30 | 2102.93 | ✗ | ✗ | ✗ |
| TMR | 65.47 | 2125.20 | ✗ | ✗ | ✗ |
| GTG | 62.84 | 827.97 | ✗ | ✗ | ✗ |
| TMC | 62.13 | 1038.88 | ✗ | ✗ | ✗ |
| LOO | 12.12 | 23.09 | 104.99 | 203.82 | 2027.16 |
| Solo | 12.78 | 22.67 | 104.20 | 202.86 | 2001.01 |
| DuoShapley | 23.25 | 44.60 | 210.28 | 417.42 | 4070.84 |

Table 2 reports the per-round runtime as the number of users increases. The results show that DuoShapley maintains clear linear scaling and remains practical up to 1000 users, while more expensive baselines such as MR, TMR, GTG, and TMC quickly become impractical even at moderate user counts. This directly supports Theorem 1 and our scalability claim in a larger-scale setting.

Table 3: **Robust user selection in the presence of noisy users on ImageNet-100.** Test accuracy (%) comparison across different data distributions. Experiments are conducted with 20% noisy users, injected with noise sampled from a Gaussian distribution with zero-mean and standard deviation $\sigma = 0.15$ (i.e., noise level). In each round, the server selects the top 20% of users based on the ranking of their estimated contributions. Every 5 rounds, the server queries updates from all users to update their contribution estimates.

| Method | Dir(10) | Dir(1) | Dir(0.1) | Dir(0.05) |
|---|---|---|---|---|
| Random | 79.37±0.41 | 78.96±0.32 | 76.61±0.24 | 74.76±0.49 |
| LOO | 76.06±2.40 | 80.24±2.08 | 77.73±2.63 | 76.50±1.27 |
| Solo | 85.52±0.12 | **85.66±0.14** | 83.76±0.17 | 80.35±0.13 |
| **DuoShapley** | **85.57±0.13** | 85.55±0.29 | **84.09±0.29** | **81.22±1.65** |

Table 3 reports robust user selection accuracy in the large-scale ImageNet-100 setting with 100 users, where 20% noisy users are injected using Gaussian noise ($\sigma = 0.15$). We consider multiple Dirichlet partitions covering a range of heterogeneity levels, from highly IID in Dir(10) to highly non-IID in Dir(0.05), with Dir(1) and Dir(0.1) lying in between. In each round, the server selects the top 20% of users according to the estimated contribution scores and refreshes these scores every 5 rounds. The results show that DuoShapley remains strong across all heterogeneity levels by combining the strengths of Solo and LOO. In near-IID settings, DuoShapley maintains high accuracy comparable to Solo, indicating robustness to noisy users, whereas LOO is more easily affected. As heterogeneity increases, DuoShapley also benefits from the complementary coalition information captured by LOO in non-IID regimes. As a result, DuoShapley achieves stronger overall accuracy while remaining both scalable and effective for robust user selection in larger and more realistic FL scenarios.

As shown in Table 4, the $\tau$-sensitivity study illustrates how $\tau$ behaves across different heterogeneity regimes and helps explain why different experiments may prefer different $\tau$ values. In the highly IID setting (Dir(10)), performance is highly stable across all tested $\tau$ values, remaining around 85%, which indicates low sensitivity

Table 4: **Sensitivity of DuoShapley to the parameter $\tau$ on ImageNet-100.** Test accuracy (%) under two representative data distributions, where Dir(10) is highly IID and Dir(0.05) is highly non-IID.

| Distribution | $\tau = 0.1$ | $\tau = 0.3$ | $\tau = 0.5$ | $\tau = 0.7$ | $\tau = 1.0$ |
|---|---|---|---|---|---|
| Dir(0.05) | 80.44±0.31 | 80.94±0.12 | 81.22±1.65 | 81.03±0.94 | 78.75±2.29 |
| Dir(10) | 85.71±0.09 | 85.57±0.09 | 85.57±0.13 | 85.81±0.18 | 85.56±0.28 |

to $\tau$. In the highly non-IID setting (Dir(0.05)), the sensitivity is more noticeable: performance is strongest for moderate $\tau$ values (0.3–0.7) and drops at $\tau = 1.0$. Intuitively, increasing $\tau$ makes the adaptive weighting more gradual and shifts more weight toward LOO; in heterogeneous settings, this makes the estimate more influenced by the LOO and therefore more vulnerable to noisy or less reliable users. These results clarify that $\tau$ was selected empirically based on validation performance. In particular, the values used in our main paper, such as $\tau = 0.3$ and $\tau = 1.0$, were chosen for their respective tasks and datasets based on validation behavior, and Table 4 now provides a practical guideline: DuoShapley is robust in IID settings, while in more heterogeneous settings a moderate $\tau$ is a safer default and should be tuned first.

## C  Proofs

### C.1  Proof of Theorem 2

By definition,

$$\phi_i^{(t)} = \alpha^{(t)}\phi_i^{\text{Solo},(t)} + (1 - \alpha^{(t)})\phi_i^{\text{LOO},(t)}. \tag{28}$$

Subtracting $\phi_i^{\star,(t)}$ gives

$$\phi_i^{(t)} - \phi_i^{\star,(t)} = \alpha^{(t)}\phi_i^{\text{Solo},(t)} + (1 - \alpha^{(t)})\phi_i^{\text{LOO},(t)} \tag{29}$$

$$- \left(\alpha_i^{\star,(t)}\phi_i^{\text{Solo},(t)} + (1 - \alpha_i^{\star,(t)})\phi_i^{\text{LOO},(t)} + r_i^{(t)}\right) \tag{30}$$

$$= (\alpha^{(t)} - \alpha_i^{\star,(t)})\left(\phi_i^{\text{Solo},(t)} - \phi_i^{\text{LOO},(t)}\right) - r_i^{(t)}. \tag{31}$$

Taking absolute values on both sides gives

$$\left|\phi_i^{(t)} - \phi_i^{\star,(t)}\right| = \left|(\alpha^{(t)} - \alpha_i^{\star,(t)})\left(\phi_i^{\text{Solo},(t)} - \phi_i^{\text{LOO},(t)}\right) - r_i^{(t)}\right|. \tag{32}$$

Applying the triangle inequality yields

$$\left|\phi_i^{(t)} - \phi_i^{\star,(t)}\right| \leq \left|(\alpha^{(t)} - \alpha_i^{\star,(t)})\left(\phi_i^{\text{Solo},(t)} - \phi_i^{\text{LOO},(t)}\right)\right| + \left|r_i^{(t)}\right|. \tag{33}$$

Finally, applying the product rule for absolute values, we obtain

$$\left|\phi_i^{(t)} - \phi_i^{\star,(t)}\right| \leq \left|\alpha^{(t)} - \alpha_i^{\star,(t)}\right|\left|\phi_i^{\text{Solo},(t)} - \phi_i^{\text{LOO},(t)}\right| + \left|r_i^{(t)}\right|. \tag{34}$$

### C.2  Proof of Theorem 3

Since

$$\varnothing \subseteq Q \subseteq \mathcal{K} \setminus \{i\}, \tag{35}$$

submodularity implies

$$\Delta\mathcal{V}_i^{(t)}(\varnothing) \geq \Delta\mathcal{V}_i^{(t)}(Q) \geq \Delta\mathcal{V}_i^{(t)}(\mathcal{K} \setminus \{i\}). \tag{36}$$

By definition,

$$\Delta \mathcal{V}_i^{(t)}(\varnothing) = \phi_i^{\text{Solo},(t)}, \qquad \Delta \mathcal{V}_i^{(t)}(\mathcal{K} \setminus \{i\}) = \phi_i^{\text{LOO},(t)}. \tag{37}$$

Hence

$$\phi_i^{\text{LOO},(t)} \leq \Delta \mathcal{V}_i^{(t)}(Q) \leq \phi_i^{\text{Solo},(t)}. \tag{38}$$

Now recall that the exact Shapley value is a convex combination of the marginal contributions $\Delta \mathcal{V}_i^{(t)}(Q)$ over all coalitions $Q \subseteq \mathcal{K} \setminus \{i\}$, with nonnegative weights summing to one. Therefore, because every term in that convex combination lies in the interval

$$\left[ \phi_i^{\text{LOO},(t)}, \phi_i^{\text{Solo},(t)} \right], \tag{39}$$

the exact Shapley value must also lie in the same interval:

$$\phi_i^{\text{LOO},(t)} \leq \phi_i^{\star,(t)} \leq \phi_i^{\text{Solo},(t)}. \tag{40}$$

### C.3  Proof of Corollary 1

By Theorem 3,

$$\phi_i^{\star,(t)} \in \left[ \phi_i^{\text{LOO},(t)}, \phi_i^{\text{Solo},(t)} \right]. \tag{41}$$

Therefore, since $\phi_i^{\star,(t)}$ lies between $\phi_i^{\text{LOO},(t)}$ and $\phi_i^{\text{Solo},(t)}$, there exists $\alpha_i^{\star,(t)} \in [0,1]$ such that

$$\phi_i^{\star,(t)} = \alpha_i^{\star,(t)} \phi_i^{\text{Solo},(t)} + (1 - \alpha_i^{\star,(t)}) \phi_i^{\text{LOO},(t)}. \tag{42}$$

Solving for $\alpha_i^{\star,(t)}$ yields

$$\alpha_i^{\star,(t)} = \frac{\phi_i^{\star,(t)} - \phi_i^{\text{LOO},(t)}}{\phi_i^{\text{Solo},(t)} - \phi_i^{\text{LOO},(t)}}. \tag{43}$$

Substituting this representation into the definition of $r_i^{(t)}$ yields $r_i^{(t)} = 0$.

