# OpenReview forum: "DuoShapley: Adaptive and Scalable Shapley Value Approximation for Federated Learning"
_TMLR — Accepted by TMLR_

### Review · Reviewer_PEmC · 2026-02-07

**Summary Of Contributions:**

The authors propose DuoShapley, an efficient and adaptive Shapley value (SV) approximation method specifically designed for large-scale Federated Learning (FL). The primary contribution is the development of an adaptive weighting scheme that balances two complementary low-order evaluations: Solo, which captures individual standalone contributions, and Leave-One-Out (LOO), which measures marginal contributions relative to the full coalition. By utilizing the cosine similarity between local and global model updates to dynamically adjust these weights, DuoShapley achieves high estimation fidelity across diverse data distributions, including both IID and highly non-IID scenarios, without requiring prior knowledge of data heterogeneity. Furthermore, the method demonstrates superior scalability with a linear complexity $O(I)$ relative to the number of users, delivering over a 200x speedup in per-round runtime compared to existing baselines like GTG and TMC.

**Audience:**

Yes

**Audience Explanation:**

the findings of this paper would interest a significant portion of TMLR's audience, particularly those focused on distributed machine learning, algorithmic efficiency, and data valuation.

**Claims And Evidence:**

Yes

**Claims Explanation:**

The claim of superior scalability is supported by concrete runtime metrics in Table 1, which demonstrate that DuoShapley’s per-round runtime grows linearly ($O(I)$) with the number of users, achieving a 200x speedup at a scale of 50 users compared to baselines like GTG and TMC. Evidence for estimation fidelity is provided through accuracy-vs-runtime trade-off analyses, where Figures 3a and 3b show DuoShapley consistently matching or exceeding the performance of the best individual order—Solo for IID and LOO for non-IID settings.

**Requested Changes:**

- The submission lacks a formal theoretical proof establishing an error bound between the estimated DuoShapley values and the exact Shapley values. The evidence for estimation fidelity relies primarily on indirect empirical validation—such as cosine distance metrics—conducted on relatively small-scale datasets and limited client pools.
- The rationale for restricting the approximation to only Solo (Order-1) and LOO (Order-n) remains insufficiently justified. While the authors provide some intuition regarding computational efficiency and the complementary nature of these extremes, the paper does not exhaustively explain why intermediate subset coalitions are entirely bypassed or how much information is lost by doing so.
- The evaluation is confined to relatively small-scale benchmarks, namely CIFAR-10 and Fashion-MNIST. Testing on more complex, large-scale datasets like ImageNet would better demonstrate the method's real-world scalability. Furthermore, certain technical concepts—such as the specific parameters of the Dirichlet distribution used for non-IID modeling—and model architectures are not explicitly detailed, which hinders the paper's self-containment.

---

> ### Author Response · Authors · 2026-03-28
> **Response to Reviewer PEmC**
>
> We thank the Reviewer PEmC for these helpful comments. To respond to the requested changes, we now add both formal theory and larger scale experiments. **Please see the Common Response above for details** and a summary below.
>
> First, the new Theorem 2 provides an explicit approximation analysis between DuoShapley and the exact Shapley value by decomposing the error into a weighting error and a residual error. This makes clear both the cost of estimating the interpolation coefficient and the approximation gap associated with skipping intermediate coalition orders. This decomposition provides an error bound relative to the exact Shapley value.
>
> Second, Theorem 3 and Corollary 1 demonstrate our intuition of why the DuoShapley design focuses on the two endpoint coalition orders, Solo and LOO. Theorem 3 shows that, under monotone submodularity, all intermediate marginal contributions are bounded by these two coalition orders, and Corollary 1 shows that under the same condition the exact Shapley value lies exactly on the interpolation between them. This gives a direct theoretical explanation for why the two endpoint coalition orders can be sufficient under mild structure, rather than treating the choice purely as a computational shortcut.
>
> Third, on the experimental side, Table A and Table B report new larger scale results on ImageNet-100 with 100 clients and a pretrained ResNet-18, which validate DuoShapley in a more practical setting. Table A shows that the method remains computationally practical at larger scale, while Table B shows that it continues to provide strong user selection performance in that setting. Together, the newly added theoretical analysis and experiments directly address the concerns on formal approximation analysis, the role of intermediate coalitions, and larger scale settings.
>
> Furthermore, for non-IID partitioning, we use the standard class-wise Dirichlet split with concentration parameter $\\gamma$: for each class, samples are distributed across clients according to a symmetric Dirichlet distribution. Smaller $\\gamma$ produces more heterogeneous splits, while larger $\\gamma$ gives more IID-like partitions. In our experiments, this includes a range of heterogeneity levels, from highly IID in $\\mathrm{Dir}(10)$ to highly non-IID in $\\mathrm{Dir}(0.05)$, with $\\mathrm{Dir}(1)$ and $\\mathrm{Dir}(0.1)$ lying in between. For model architectures, the ImageNet-100 experiments use a pretrained ResNet-18, where the final fully connected layer is replaced by a 100-class classifier, and only the classifier is fine-tuned. For CIFAR-10 and Fashion-MNIST, we use convolutional neural networks (CNNs) architectures consisting of convolutional blocks, each followed by ReLU activations and max-pooling, and a fully connected classifier. The CIFAR-10 model uses six convolutional layers, while the Fashion-MNIST model uses five convolutional layers.

---

### Review · Reviewer_W9Ux · 2026-02-23

**Summary Of Contributions:**

The paper makes several noteworthy contributions to the field of FL:

In-depth Analysis of Solo and LOO: It provides a valuable empirical analysis of two extreme but efficient coalition orders, clearly demonstrating their complementary strengths and weaknesses: Solo performs well in IID settings but underestimates valuable, unique contributors in non-IID settings, while LOO excels in non-IID settings but lacks resolution in homogeneous ones.

Introduction of DuoShapley: The authors propose a novel, efficient, and adaptive algorithm that dynamically balances Solo and LOO contributions. The use of cosine similarity between local and global updates as a proxy for data heterogeneity to guide this balance is a simple yet effective idea.

Demonstration of Practical Utility: The paper goes beyond simple contribution estimation by showcasing DuoShapley in a downstream application—robust user selection. The results convincingly show that using DuoShapley's contribution estimates to select participants leads to better model accuracy and robustness against noisy users compared to using either Solo or LOO alone.

State-of-the-Art Scalability: A key strength is the method's linear scalability. The paper provides compelling runtime comparisons (Table 1) showing that DuoShapley achieves a >200x speedup over methods like GTG and TMC in a 50-user setting, making it a practical solution for large-scale FL deployments.

**Audience:**

Yes

**Audience Explanation:**

I think that this paper is interested enough.

**Broader Impact Concerns:**

I have no broader impact concerns.

**Claims And Evidence:**

Yes

**Claims Explanation:**

The runtime results in Table 1 are stark and unequivocal. They provide clear, quantitative evidence for the claim that DuoShapley is far more scalable than existing approximation methods (MR, TMR, GTG, TMC) and maintains the linear efficiency of its base components, Solo and LOO.

The experimental setup is robust. The use of user addition (Figure 3) and removal (Appendix A.2) tasks provides a meaningful way to evaluate the quality of contribution rankings. Figure 3a and 3b clearly show DuoShapley's adaptive nature, matching the best-performing method in both IID and non-IID settings. The comparison with fixed α values in Figures 3c and 3d further validates the effectiveness of the proposed adaptive weighting strategy.

The experiments in Figure 5 are particularly strong. They systematically vary data distribution (IID to non-IID), noise level, and the proportion of noisy users. The results consistently show that DuoShapley inherits the resilience of Solo in high-noise scenarios and the sensitivity of LOO in non-IID scenarios, leading to robust performance across all tested conditions. This provides convincing evidence for its practical value.

The paper is well-written and structured. The motivation is clearly explained with the help of Figure 2. The DuoShapley algorithm is presented in a clear, step-by-step manner (Algorithm 1), and the experimental methodology is described in sufficient detail (Section 5.1).

**Requested Changes:**

The role of $\tau$ is explained, but its sensitivity is not deeply explored. How was τ tuned for the different experiments (e.g., 0.3 for user addition and 1.0 for user selection in the 50-user setting)? Is the method sensitive to this parameter, and can any guidelines be provided for setting it in practice?

While the method is efficient, it still requires evaluating the utility on a validation set for both Solo and LOO models (lines 16 and 18 in Algorithm 1). This cost, though linear, could be discussed. Future directions could include exploring other similarity metrics beyond cosine, or applying the DuoShapley principle to other FL paradigms like personalized FL or vertical FL.

To better position the work within the current state of the art, the authors should consider citing recent papers that address related challenges in FL, such as task offloading with privacy and robustness against attacks and heterogeneity. The following papers are recommended:

M. Min et al. "Task Offloading with Differential Privacy in Multi-Access Edge Computing: An A3C-Based Approach." IEEE Transactions on Cognitive Communications and Networking, 2026. (This work is relevant as it deals with optimizing participation and privacy in a distributed system, which shares goals with robust user selection in FL).

S. Zuo et al. "Federated learning resilient to byzantine attacks and data heterogeneity." IEEE Transactions on Mobile Computing, 2025. (This paper is directly relevant as it tackles both data heterogeneity and robustness to malicious participants, which are central themes in the DuoShapley paper, particularly in the user selection application).

---

> ### Author Response · Authors · 2026-03-28
> **Response to Reviewer W9Ux**
>
> We thank the Reviewer W9Ux for these helpful suggestions. In addition to the **Common Response provided above**, below are responses to this reviewer's specific comments.
>
> First, we add a dedicated $\\tau$-sensitivity study in Table C using the larger-scale ImageNet-100 setting with more clients and a pretrained ResNet-18 under classifier fine-tuning. This additional study illustrates how $\\tau$ behaves across different heterogeneity regimes and helps explain why different experiments may prefer different $\\tau$ values. In the highly IID setting ($\\mathrm{Dir}(10)$), performance is highly stable across all tested $\\tau$ values, remaining around 85%, which indicates low sensitivity to $\\tau$. In the highly non-IID setting ($\\mathrm{Dir}(0.05)$), the sensitivity is more noticeable: performance is strongest for moderate $\\tau$ values ($0.3$--$0.7$) and drops at $\\tau=1.0$. Intuitively, increasing $\\tau$ makes the adaptive weighting sharper and can place excessive emphasis on one endpoint; in heterogeneous settings, this makes the estimate more influenced by the LOO and therefore more vulnerable to noisy or less reliable users. We will clarify that $\\tau$ was selected empirically based on validation performance. In particular, the values used in our paper, such as $\\tau=0.3$ and $\\tau=1.0$, were chosen for their respective tasks and datasets based on validation behavior, and Table C now provides a practical guideline: DuoShapley is robust in IID settings, while in more heterogeneous settings a moderate $\\tau$ is a safer default and should be tuned first.
>
> &nbsp;
>
> **Table C.** Sensitivity of DuoShapley to the parameter $\\tau$ on ImageNet-100. Test accuracy (%) under two representative data distributions, where $\\mathrm{Dir}(10)$ is highly IID and $\\mathrm{Dir}(0.05)$ is highly non-IID.
>
> | &nbsp;Distribution&nbsp; | &nbsp;$\\tau=0.1$&nbsp; | &nbsp;$\\tau=0.3$&nbsp; | &nbsp;$\\tau=0.5$&nbsp; | &nbsp;$\\tau=0.7$&nbsp; | &nbsp;$\\tau=1.0$&nbsp; |
> |---|---:|---:|---:|---:|---:|
> | &nbsp;Dir(0.05)&nbsp; | 80.44±0.31 | 80.94±0.12 | 81.22±1.65 | 81.03±0.94 | 78.75±2.29 |
> | &nbsp;Dir(10)&nbsp; | 85.71±0.09 | 85.57±0.09 | 85.57±0.13 | 85.81±0.18 | 85.56±0.28 |
>
> &nbsp;
>
> Second, although DuoShapley still evaluates validation utility for both Solo and LOO, its per-round complexity remains linear in the number of users, which we now formalize in Theorem 1. We further support this with new larger-scale results in Tables A and B, where we validate DuoShapley in broader and more practical settings, showing that the method remains scalable and effective beyond the original small-scale benchmarks. **Please see Common Response above for details.**
>
> Third, we thank the reviewer for suggesting these related works, and we will include them in the revised paper to better position our contribution relative to recent work on privacy, robustness, and heterogeneity in federated learning.

---

### Review · Reviewer_mqWr · 2026-03-15

**Summary Of Contributions:**

This paper investigates the problem of efficiently estimating user contributions in federated learning (FL) through Shapley value (SV) approximation. It is known that the exact computation of SV regarding contribution attribution is intractable due to exponential complexity in the number of participants. The authors report that two efficient coalition orders, including Solo (order-1) and Leave-One-Out (LOO), may exhibit complementary strengths depending on the level of data heterogeneity. Solo tends to perform better under IID settings, while LOO is more informative under non-IID distributions. With these two coalition orders, DuoShapley is proposed to adaptively conduct the approximation by combining Solo and LOO estimates. Experimental results demonstrate that the proposed method achieves competitive accuracy while maintaining strong scalability in federated learning systems. The followings are the strengths of this paper.

Strengths
1. The problem tackled in this paper, i.e., how to evaluate user contributions efficiently in large-scale decentralized systems, is an important and practical problem in federated learning systems.
2. The proposed method is simple and intuitive. Combining two computationally efficient coalition orders with an adaptive weighting strategy allows the proposed method to preserve a linear computational complexity while improving robustness across varying data heterogeneity levels.
3. The empirical observations of Solo and LOO are well supported by experiments.
4. The experiments are comprehensive to demonstrate the effectiveness of the proposed method.

**Additional Comments:**

NIL.

**Audience:**

Yes

**Audience Explanation:**

The problem tackled in this paper, i.e., how to evaluate user contributions efficiently in large-scale decentralized systems, is an important and practical problem in federated learning systems.

**Claims And Evidence:**

No

**Claims Explanation:**

Most claims made in the submission are supported by empirical results. A solid, theoretical analysis would be very helpful to improve the overall contributions of the paper. Details are listed under "Requested Changes."

**Requested Changes:**

The authors are strongly encouraged to address the following weaknesses found in the paper.

Weaknesses
1. The novelty of this paper is somewhat incremental. The combination of Solo and LOO contributions through a similarity-based approach may be seen as an engineering refinement rather than a fundamentally new SV approximation framework. The novelty and contributions of this paper can be significantly improved if deeper theoretical insights explaining why the proposed weighting scheme approximates Shapley values more accurately could be provided.
2. Besides, this paper lacks theoretical analysis regarding approximation guarantees or error bounds relative to the true Shapley values. The empirical results provided in this paper are comprehensive, but a stronger theoretical analysis may further strengthen the contributions.
3. Evaluation of SV values is mainly conducted under small-scale settings. This makes it difficult to assess the approximation accuracy under more realistic settings.

---

> ### Author Response · Authors · 2026-03-28
> **Response to Reviewer mqWr**
>
> We thank the Reviewer mqWr for this helpful feedback. We have now added both new theory and larger scale experiments. **Please see the Common Response above for details.** For convenience, we reiterate the main points specifically to address this reviewer's concerns.
>
> On the theory side, Theorem 1 formalizes the linear complexity of DuoShapley, Theorem 2 gives an explicit error decomposition with respect to the exact Shapley value, and Theorem 3 together with Corollary 1 explains when focusing only on Solo and LOO is justified. In particular, Theorem 2 separates the approximation error into a weighting error and a residual error, while Theorem 3 and Corollary 1 show that, under monotone submodularity, the residual term vanishes because the exact Shapley value lies exactly on the interpolation between Solo and LOO. This justifies the design of DuoShapley as a combination of Solo and LOO: it approximates the exact Shapley value through interpolation between them and comes with an explicit description of the approximation error.
>
> On the experimental side, Table A reports the larger scale runtime results on ImageNet-100, and Table B reports robust user selection performance in the same larger scale setting. Table A supports the scalability claim by showing that DuoShapley remains practical as the number of users grows, while Table B shows that DuoShapley continues to identify strong users effectively and achieves strong accuracy in this larger-scale setting.
>
> We hope that these new theoretical and experimental results address the reviewer's concerns on theoretical justification, approximation error, and larger scale evaluation.

---

### Author Response · Authors · 2026-03-28
**Response to Common Comments of Reviewers**

Thank you to all reviewers for the careful reading and constructive feedback. We provide responses to each reviewer individually.

In addition, across the three reviews, two common concerns emerged: (1) the need for stronger theoretical analysis of DuoShapley; and (2) the need for larger-scale experiments. To address these common concerns, we have added new results in both directions:

On the theory side, we now provide theoretical support for DuoShapley from three angles. We first formalize its linear-time complexity, which is the main reason for restricting attention to Solo and LOO. We then derive an error decomposition with respect to the exact Shapley value, showing that the approximation error consists of two parts: a weighting error from estimating the interpolation coefficient, and a residual error that captures the approximation gap introduced by skipping intermediate coalition orders. Finally, under a monotone submodular utility model, we show that Solo and LOO become principled endpoints that bound all intermediate marginal contributions. Under monotone submodularity, the exact Shapley value lies on the interpolation of Solo and LOO, so the residual vanishes and the remaining error comes only from estimating the interpolation coefficient.

On the experimental side, we run new experiments on the ImageNet-100 dataset with 100 clients and a pretrained ResNet-18, including runtime and robust user selection results. These additions strengthen both the theoretical grounding and the empirical realism of the paper. If the paper is accepted, we can include these additional results in the final version.

---

> ### Author Response · Authors · 2026-03-28
> **Common Response: Theoretical Analysis**
>
> &nbsp;
> ## 1. Theoretical Analysis
> &nbsp;
> ### 1.1 Preliminaries
>
> Here, we provide a theoretical analysis of DuoShapley. We first show that the method has linear complexity. We then derive an error decomposition with respect to the exact Shapley value. Here, the exact Shapley value denotes the full coalition-based Shapley value at round $t$, namely, the average marginal contribution of user $i$ over all possible coalitions under the utility $\\mathcal{V}^{(t)}$. Finally, under a monotone submodular utility model, we show that Solo and LOO form two principled extreme coalition orders that bound the exact Shapley value and motivate its approximation through an interpolation coefficient.
>
> Let $\\mathcal{K}$ denote the user set with $|\\mathcal{K}|=I$. For a fixed round $t$, let $\\mathcal{V}^{(t)}:2^\\mathcal{K}\\to\\mathbb{R}$ denote the coalition utility evaluated on the server-side validation set. For any user $i\\in \\mathcal{K}$ and coalition $Q\\subseteq \\mathcal{K}\\setminus\\{i\\}$, define the marginal contribution
>
> $$
> \\Delta \\mathcal{V}_i^{(t)}(Q):=\\mathcal{V}^{(t)}(Q\\cup\\{i\\})-\\mathcal{V}^{(t)}(Q).
> $$
>
> The exact Shapley value of user $i$ at round $t$ is
>
> $$
> \\phi_i^{\\star,(t)}:=\\sum_{Q\\subseteq \\mathcal{K}\\setminus\\{i\\}} \\frac{|Q|!\\,(I-|Q|-1)!}{I!}\\, \\Delta \\mathcal{V}_i^{(t)}(Q),
> $$
>
> i.e., the weighted average of user $i$'s marginal contributions over all coalition orders [R1].
>
> We recall the two endpoints used by DuoShapley:
>
> $$
> \\phi_i^{\\mathrm{Solo},(t)} := \\Delta \\mathcal{V}_i^{(t)}(\\varnothing)=\\mathcal{V}^{(t)}(\\{i\\})-\\mathcal{V}^{(t)}(\\varnothing),
> $$
>
> $$
> \\phi_i^{\\mathrm{LOO},(t)} := \\Delta \\mathcal{V}_i^{(t)}(\\mathcal{K}\\setminus\\{i\\})=\\mathcal{V}^{(t)}(\\mathcal{K})-\\mathcal{V}^{(t)}(\\mathcal{K}\\setminus\\{i\\}),
> $$
>
> and the DuoShapley estimate
>
> $$
> \\phi_i^{(t)}=\\alpha^{(t)}\\phi_i^{\\mathrm{Solo},(t)}+(1-\\alpha^{(t)})\\phi_i^{\\mathrm{LOO},(t)}, \\qquad \\alpha^{(t)}\\in[0,1].
> $$
>
> &nbsp;
> ### 1.2 Linear Complexity
>
> We first formalize the computational advantage of DuoShapley. Since DuoShapley only evaluates the two endpoint coalition orders, Solo and LOO, its cost grows linearly with the number of users.
>
> **Theorem 1 (Linear complexity).** *For a fixed round $t$, let $C_V$ denote the cost of one utility evaluation on the validation set, and let $p$ denote the model dimension. DuoShapley requires $O(I)$ utility evaluations and $O(Ip)$ additional arithmetic operations. Hence its per-round complexity is*
>
> $$
> O(IC_V + Ip),
> $$
>
> *which is linear in the number of users.*
>
> **Proof.** For each user $i$, DuoShapley computes one Solo utility and one LOO utility, so the number of utility evaluations is $O(I)$. Since each utility evaluation costs $C_V$, this contributes $O(IC_V)$.
>
> The method also computes cosine similarity weight per user and aggregates them into a shared round level coefficient. Each similarity computation costs $O(p)$, so this contributes $O(Ip)$ overall. Combining the two terms gives
>
> $$
> O(IC_V + Ip),
> $$
>
> which is linear in the number of users.

---

> ### Author Response · Authors · 2026-03-28
> **Common Response: Theoretical Analysis**
>
> &nbsp;
>
> ### 1.3 Error Analysis
>
> Efficiency alone is not sufficient; we also need to understand what approximation error DuoShapley makes compared to the exact Shapley value. The next theorem gives a round-wise error decomposition that isolates two sources of error in our estimator and provides bounds, without any assumptions.
>
>
> **Theorem 2 (Error decomposition).** *Fix a round $t$ and user $i$. Let $\\alpha_i^{\\star,(t)}\\in[0,1]$ denote the optimal coefficient, and define the residual*
>
> $$
> r_i^{(t)}:=\\phi_i^{\\star,(t)}-(\\alpha_i^{\\star,(t)}\\phi_i^{\\mathrm{Solo},(t)}+(1-\\alpha_i^{\\star,(t)})\\phi_i^{\\mathrm{LOO},(t)}).
> $$
>
> *Then the DuoShapley estimation error satisfies*
>
> $$
> |\\phi_i^{(t)}-\\phi_i^{\\star,(t)}| \\le |r_i^{(t)}| + |\\alpha^{(t)}-\\alpha_i^{\\star,(t)}| \\, |\\phi_i^{\\mathrm{Solo},(t)}-\\phi_i^{\\mathrm{LOO},(t)}|.
> $$
>
> **Proof.** By definition,
>
> $$
> \\phi_i^{(t)}=\\alpha^{(t)}\\phi_i^{\\mathrm{Solo},(t)}+(1-\\alpha^{(t)})\\phi_i^{\\mathrm{LOO},(t)}.
> $$
>
> Subtracting $\\phi_i^{\\star,(t)}$ gives
>
> $$
> \\phi_i^{(t)}-\\phi_i^{\\star,(t)}=\\alpha^{(t)}\\phi_i^{\\mathrm{Solo},(t)}+(1-\\alpha^{(t)})\\phi_i^{\\mathrm{LOO},(t)}-(\\alpha_i^{\\star,(t)}\\phi_i^{\\mathrm{Solo},(t)}+(1-\\alpha_i^{\\star,(t)})\\phi_i^{\\mathrm{LOO},(t)}+r_i^{(t)}).
> $$
>
> Therefore,
>
> $$
> \\phi_i^{(t)}-\\phi_i^{\\star,(t)}=(\\alpha^{(t)}-\\alpha_i^{\\star,(t)})(\\phi_i^{\\mathrm{Solo},(t)}-\\phi_i^{\\mathrm{LOO},(t)})-r_i^{(t)}.
> $$
>
> Taking absolute values on both sides gives
>
> $$
> |\\phi_i^{(t)}-\\phi_i^{\\star,(t)}| = |(\\alpha^{(t)}-\\alpha_i^{\\star,(t)})(\\phi_i^{\\mathrm{Solo},(t)}-\\phi_i^{\\mathrm{LOO},(t)})-r_i^{(t)}|.
> $$
>
> Applying the triangle inequality yields
>
> $$
> |\\phi_i^{(t)}-\\phi_i^{\\star,(t)}| \\le |(\\alpha^{(t)}-\\alpha_i^{\\star,(t)})(\\phi_i^{\\mathrm{Solo},(t)}-\\phi_i^{\\mathrm{LOO},(t)})| + |r_i^{(t)}|.
> $$
>
> Finally, applying the product rule for absolute values, we obtain
>
> $$
> |\\phi_i^{(t)}-\\phi_i^{\\star,(t)}| \\le |\\alpha^{(t)}-\\alpha_i^{\\star,(t)}| \\, |\\phi_i^{\\mathrm{Solo},(t)}-\\phi_i^{\\mathrm{LOO},(t)}| + |r_i^{(t)}|.
> $$
>
> **Theorem 2** shows that DuoShapley has two distinct error components. The first is a *weighting error*, $|\\alpha^{(t)}-\\alpha_i^{\\star,(t)}| \\, |\\phi_i^{\\mathrm{Solo},(t)}-\\phi_i^{\\mathrm{LOO},(t)}|$, which reflects how far the computed coefficient $\\alpha^{(t)}$ is from the optimal coefficient $\\alpha_i^{\\star,(t)}$. The second is a *residual error*, $|r_i^{(t)}|$, which captures the approximation gap introduced by skipping intermediate coalition orders. Thus, the weighting error comes from estimating the interpolation coefficient, while the residual error comes from restricting to the two endpoint coalition orders, Solo and LOO.
>
> &nbsp;
>
> **Special Case: Monotone Submodular Utility.** Next, we ask when focusing only on Solo and LOO is well justified. To facilitate analysis, we make the mild assumption of monotone submodular utility model, where marginal contributions decrease with coalition size. We show that, in this case, Solo and LOO are the upper and lower bounds on marginal contributions.
>
> &nbsp;
>
> **Assumption 1 (Monotone submodular utility).** *For a fixed round $t$, the utility function $\\mathcal{V}^{(t)}$ is:*
>
> - ***monotone:*** $\\mathcal{V}^{(t)}(A)\\le \\mathcal{V}^{(t)}(B)$, $\\quad$ *whenever* $A\\subseteq B$;
> - ***submodular:*** *for all* $A\\subseteq B\\subseteq \\mathcal{K}\\setminus\\{i\\}$, $\\quad$ $\\Delta \\mathcal{V}_i^{(t)}(A)\\ge \\Delta \\mathcal{V}_i^{(t)}(B).$
>
> &nbsp;
>
> **Theorem 3 (Bounds on coalition marginals).** *Under Assumption 1, for every round $t$, user $i$, and coalition $Q\\subseteq \\mathcal{K}\\setminus\\{i\\}$,*
>
> $$
> \\phi_i^{\\mathrm{LOO},(t)} \\le \\Delta \\mathcal{V}_i^{(t)}(Q) \\le \\phi_i^{\\mathrm{Solo},(t)}.
> $$
>
> *Consequently,*
>
> $$
> \\phi_i^{\\mathrm{LOO},(t)} \\le \\phi_i^{\\star,(t)} \\le \\phi_i^{\\mathrm{Solo},(t)}.
> $$
>
> **Proof.** Since
>
> $$
> \\varnothing \\subseteq Q \\subseteq \\mathcal{K}\\setminus\\{i\\},
> $$
>
> submodularity implies
>
> $$
> \\Delta \\mathcal{V}_i^{(t)}(\\varnothing) \\ge \\Delta \\mathcal{V}_i^{(t)}(Q) \\ge \\Delta \\mathcal{V}_i^{(t)}(\\mathcal{K}\\setminus\\{i\\}).
> $$
>
> By definition,
>
> $$
> \\Delta \\mathcal{V}_i^{(t)}(\\varnothing)=\\phi_i^{\\mathrm{Solo},(t)}, \\qquad \\Delta \\mathcal{V}_i^{(t)}(\\mathcal{K}\\setminus\\{i\\})=\\phi_i^{\\mathrm{LOO},(t)}.
> $$
>
> Hence
>
> $$
> \\phi_i^{\\mathrm{LOO},(t)} \\le \\Delta \\mathcal{V}_i^{(t)}(Q) \\le \\phi_i^{\\mathrm{Solo},(t)}.
> $$
>
> Now recall that the Shapley value is a convex combination of the marginal contributions $\\Delta \\mathcal{V}_i^{(t)}(Q)$ over all coalitions $Q\\subseteq \\mathcal{K}\\setminus\\{i\\}$, with nonnegative weights summing to one. Therefore, because every term in that convex combination lies in the interval
>
> $$
> [\\phi_i^{\\mathrm{LOO},(t)},\\phi_i^{\\mathrm{Solo},(t)}],
> $$
>
> the exact Shapley value must also lie in the same interval:
>
> $$
> \\phi_i^{\\mathrm{LOO},(t)} \\le \\phi_i^{\\star,(t)} \\le \\phi_i^{\\mathrm{Solo},(t)}.
> $$

---

> ### Author Response · Authors · 2026-03-28
> **Common Response: Theoretical Analysis**
>
> &nbsp;
>
> **Theorem 3** demonstrates our intuition of why the DuoShapley design focuses on the two endpoint coalition orders, Solo and LOO. Under monotone submodularity, a user's marginal contribution decreases as the coalition grows, so every intermediate coalition marginal lies between these two extremes. Since the exact Shapley value is a weighted average of all such marginal contributions, it must also lie between Solo and LOO.
>
> &nbsp;
>
> **Corollary 1 (Exact interpolation between Solo and LOO).** *Under Assumption 1, if*
>
> $$
> \\phi_i^{\\mathrm{Solo},(t)}\\neq \\phi_i^{\\mathrm{LOO},(t)},
> $$
>
> *then there exists an interpolation coefficient*
>
> $$
> \\alpha_i^{\\star,(t)}=\\frac{\\phi_i^{\\star,(t)}-\\phi_i^{\\mathrm{LOO},(t)}}{\\phi_i^{\\mathrm{Solo},(t)}-\\phi_i^{\\mathrm{LOO},(t)}}\\in[0,1]
> $$
>
> *such that*
>
> $$
> \\phi_i^{\\star,(t)}=\\alpha_i^{\\star,(t)}\\phi_i^{\\mathrm{Solo},(t)}+(1-\\alpha_i^{\\star,(t)})\\phi_i^{\\mathrm{LOO},(t)}.
> $$
>
> *Hence the residual in Theorem 2 is exactly zero:*
>
> $$
> r_i^{(t)}=0.
> $$
>
> **Proof.** By Theorem 3,
>
> $$
> \\phi_i^{\\star,(t)}\\in[\\phi_i^{\\mathrm{LOO},(t)},\\phi_i^{\\mathrm{Solo},(t)}].
> $$
>
> Therefore, since $\\phi_i^{\\star,(t)}$ lies between $\\phi_i^{\\mathrm{LOO},(t)}$ and $\\phi_i^{\\mathrm{Solo},(t)}$, there exists $\\alpha_i^{\\star,(t)}\\in[0,1]$ such that
>
> $$
> \\phi_i^{\\star,(t)}=\\alpha_i^{\\star,(t)}\\phi_i^{\\mathrm{Solo},(t)}+(1-\\alpha_i^{\\star,(t)})\\phi_i^{\\mathrm{LOO},(t)}.
> $$
>
> Solving for $\\alpha_i^{\\star,(t)}$ yields
>
> $$
> \\alpha_i^{\\star,(t)}=\\frac{\\phi_i^{\\star,(t)}-\\phi_i^{\\mathrm{LOO},(t)}}{\\phi_i^{\\mathrm{Solo},(t)}-\\phi_i^{\\mathrm{LOO},(t)}}.
> $$
>
> Substituting this representation into the definition of $r_i^{(t)}$ yields $r_i^{(t)}=0$.
>
> **Corollary 1** shows that, under monotone submodularity, the exact Shapley value can be written exactly as a weighted average of Solo and LOO. Thus, ignoring intermediate coalition orders does not introduce a separate representation error; the only remaining approximation error comes from how well the adaptive weight $\\alpha^{(t)}$ matches the optimal coefficient $\\alpha_i^{\\star,(t)}$. In the main paper, we compute $\\alpha^{(t)}$ using the cosine similarity based method in Equations (6) to (8), which uses the alignment between local and global updates for balancing Solo and LOO. Stronger alignment favors Solo, while weaker alignment points to stronger complementarity and therefore favors LOO; see Section 4.2 of the main paper for additional details.
>
> **Scope of assumptions.** The monotone submodular analysis above justifies the design of DuoShapley from a theoretical perspective. In practical federated learning, validation accuracy may not be strictly monotone or submodular due to optimization noise, label noise, or noisy participants. Theorem 2 holds without additional assumptions on the utility function and therefore applies generally. Theorem 3 shows that, under monotone submodularity, Solo and LOO bound the intermediate marginal contributions. Corollary 1 further shows that the exact Shapley value can be written exactly as an interpolation between Solo and LOO.

---

> ### Author Response · Authors · 2026-03-28
> **Common Response: Experiments**
>
> &nbsp;
>
> ## 2. Experiments
>
> To address the reviewers’ requests for evaluation in a larger setting, we add new experiments on the ImageNet-100 dataset [R2, R3], which is a 100-class subset of ImageNet [R4]. Compared with CIFAR-10 and Fashion-MNIST used in our main paper, ImageNet-100 contains 100 classes, higher-resolution images, and more data samples, making it a larger-scale and more practical benchmark. We also evaluate a larger-scale setting with 100 clients and a pretrained model ResNet-18 [R5] under classifier fine-tuning, extending beyond the scale of the existing experiments in the paper.
>
> **Table A** reports the per-round runtime as the number of users increases. The results show that DuoShapley maintains clear linear scaling and remains practical up to 1000 users, while more expensive baselines such as MR, TMR, GTG, and TMC quickly become impractical even at moderate user counts. This directly supports Theorem 1 and our scalability claim in a larger-scale setting.
>
> **Table B** reports robust user selection accuracy in the large-scale ImageNet-100 setting with 100 clients, where 20% noisy users are injected using Gaussian noise ($\\sigma=0.15$). We consider multiple Dirichlet partitions covering a range of heterogeneity levels, from highly IID in $\\mathrm{Dir}(10)$ to highly non-IID in $\\mathrm{Dir}(0.05)$, with $\\mathrm{Dir}(1)$ and $\\mathrm{Dir}(0.1)$ lying in between. In each round, the server selects the top 20% of users according to the estimated contribution scores and refreshes these scores every 5 rounds. The results show that DuoShapley remains strong across all heterogeneity levels by combining the strengths of Solo and LOO. In near-IID settings, DuoShapley maintains high accuracy comparable to Solo, indicating robustness to noisy users, whereas LOO is more easily affected. As heterogeneity increases, DuoShapley also benefits from the complementary coalition information captured by LOO in non-IID regimes. As a result, DuoShapley achieves stronger overall accuracy while remaining both scalable and effective for robust user selection in larger and more realistic FL scenarios.
>
> &nbsp;
>
> **Table A.** Per-round runtime (in seconds) of each method for varying numbers of users on ImageNet-100. All methods use FedAvg-style aggregation.
>
> | &nbsp;Method&nbsp; | &nbsp;5 Users&nbsp; | &nbsp;10 Users&nbsp; | &nbsp;50 Users&nbsp; | &nbsp;100 Users&nbsp; | &nbsp;1000 Users&nbsp; |
> |---|---:|---:|---:|---:|---:|
> | &nbsp;MR&nbsp; | 64.30 | 2102.93 | ✗ | ✗ | ✗ |
> | &nbsp;TMR&nbsp; | 65.47 | 2125.20 | ✗ | ✗ | ✗ |
> | &nbsp;GTG&nbsp; | 62.84 | 827.97 | ✗ | ✗ | ✗ |
> | &nbsp;TMC&nbsp; | 62.13 | 1038.88 | ✗ | ✗ | ✗ |
> | &nbsp;LOO&nbsp; | 12.12 | 23.09 | 104.99 | 203.82 | 2027.16 |
> | &nbsp;Solo&nbsp; | 12.78 | 22.67 | 104.20 | 202.86 | 2001.01 |
> | &nbsp;DuoShapley&nbsp; | 23.25 | 44.60 | 210.28 | 417.42 | 4070.84 |
>
> &nbsp;
>
> **Table B.** Robust user selection in the presence of noisy users on ImageNet-100. Test accuracy (%) comparison across different data distributions. Experiments are conducted with 20% noisy users, injected with noise sampled from a Gaussian distribution with zero-mean and standard deviation $\\sigma=0.15$ (i.e., noise level). In each round, the server selects the top 20% of users based on the ranking of their estimated contributions. Every 5 rounds, the server queries updates from all users to update their contribution estimates.
>
> | &nbsp;Method&nbsp; | &nbsp;Dir(10)&nbsp; | &nbsp;Dir(1)&nbsp; | &nbsp;Dir(0.1)&nbsp; | &nbsp;Dir(0.05)&nbsp; |
> |---|---:|---:|---:|---:|
> | &nbsp;Random&nbsp; | 79.37±0.41 | 78.96±0.32 | 76.61±0.24 | 74.76±0.49 |
> | &nbsp;LOO&nbsp; | 76.06±2.40 | 80.24±2.08 | 77.73±2.63 | 76.50±1.27 |
> | &nbsp;Solo&nbsp; | 85.52±0.12 | **85.66±0.14** | 83.76±0.17 | 80.35±0.13 |
> | **&nbsp;DuoShapley&nbsp;** | **85.57±0.13** | 85.55±0.29 | **84.09±0.29** | **81.22±1.65** |
>
> &nbsp;
>
> ## References
>
> [R1] Amirata Ghorbani and James Zou. *Data Shapley: Equitable Valuation of Data for Machine Learning*. In International Conference on Machine Learning, pages 2242–2251. PMLR, 2019.
>
> [R2] *ImageNet-100 Dataset Repository*. https://huggingface.co/datasets/clane9/imagenet-100, 2024.
>
> [R3] Yonglong Tian, Dilip Krishnan, and Phillip Isola. *Contrastive Multiview Coding*. In European Conference on Computer Vision, pages 776–794. Springer, 2020.
>
> [R4] Jia Deng, Wei Dong, Richard Socher, Li-Jia Li, Kai Li, and Li Fei-Fei. *ImageNet: A Large-Scale Hierarchical Image Database*. In 2009 IEEE Conference on Computer Vision and Pattern Recognition, pages 248–255. IEEE, 2009.
>
> [R5] Kaiming He, Xiangyu Zhang, Shaoqing Ren, and Jian Sun. *Deep Residual Learning for Image Recognition*. In Proceedings of the IEEE Conference on Computer Vision and Pattern Recognition, pages 770–778, 2016.

---

### Decision · Action_Editor_AmHt · 2026-06-03

**Recommendation:** Accept with minor revision

**Additional Comments:**

- Please incorporate the additional theoretical analysis into the manuscript
- Please correct the vast majority of the citations using \citep
- Please precisely define Solo and LOO with technical details in Sec 3 before motivating the new DuoShapley in Sec 4

**Audience:**

Yes

**Audience Explanation:**

Federated learning is an emerging learning paradigm for collaborative learning without compromising data privacy

**Claims And Evidence:**

Yes

**Claims Explanation:**

This paper addresses user valuation in federated learning by introducing DuoShapley, which scores a user's contribution by combining their standalone value ("Solo") with their value relative to the whole group ("Leave-One-Out"). The paper demonstrates the effectiveness and scalability of the proposed methods with solid experimental results. During the rebuttal, the authors provided additional theoretical insights into their algorithm.